# Similarity and Matching of Neural Network Representations

**Adrián Csiszárik[†,°], Péter Kőrösi-Szabó[†], Ákos K. Matszangosz[†]**
**Gergely Papp[†], Dániel Varga[†]**

[†]Alfréd Rényi Insititute of Mathematics, Budapest, Hungary
[°]Eötvös Loránd University, Budapest, Hungary
{csadrian, koszpe, matszang, gergopool, daniel}@renyi.hu

## Abstract

We employ a toolset — dubbed Dr. Frankenstein — to analyse the similarity of representations in deep neural networks. With this toolset, we aim to match the activations on given layers of two trained neural networks by joining them with a stitching layer. We demonstrate that the inner representations emerging in deep convolutional neural networks with the same architecture but different initializations can be matched with a surprisingly high degree of accuracy even with a single, affine stitching layer. We choose the stitching layer from several possible classes of linear transformations and investigate their performance and properties. The task of matching representations is closely related to notions of similarity. Using this toolset, we also provide a novel viewpoint on the current line of research regarding similarity indices of neural network representations: the perspective of the performance on a task.

## 1 Introduction

A central topic in the analysis of deep neural networks is the investigation of the learned representations. A good understanding of the features learned on inner layers provides means to analyse and advance deep learning systems. A fundamental question in this line of inquiry is *"when are two representations similar?"* We contribute to this line of research by introducing a novel approach to study the above question. The key idea of our work is to ask the following, somewhat different question: *"in what way are two representations similar, once we know that they are similar?"* In this paper, we present a conceptual framework and a methodology that allows to meaningfully pose and study this question. For this purpose, we will distinguish two approaches to define similarity: *representational similarity* and *functional similarity*.

**Representational similarity**    Representational similarity notions define similarity via computing statistics on two different data embeddings that capture appropriate geometrical or statistical properties. There exist several statistical measures, [Raghu et al., 2017, Morcos et al., 2018, Kornblith et al., 2019] each of which serves as a different notion of similarity. These measures have been used successfully to obtain valuable insights about the inner workings of deep neural networks [Nguyen et al., 2021, Mirzadeh et al., 2021, Neyshabur et al., 2020, Wu et al., 2020].

**Functional similarity**    While representational similarity concerns the data embeddings, in contrast, functional similarity concerns the functions that produce and process these embeddings, i.e., parts of the function compositions that the neural networks realize. With these elements, we can ask new types of questions that we were not able to ask using solely the data embeddings. For example, we can ask the following: *"can network B achieve its task using the representations of network A?"*

35th Conference on Neural Information Processing Systems (NeurIPS 2021).

Our paper focuses on this specific theme. In particular, we investigate this by taking the activations of network A at a given layer, transform it with an affine map, then use the result as an input of the same layer in network B. In other words, we stitch together the two networks with an affine stitching layer. This technique first appeared in Lenc and Vedaldi [2019] to study the invariance and equivariance properties of convolutional networks. Evaluating the performance of this combined network provides an alternative viewpoint on similarity of representations, the viewpoint of functional similarity.

We investigate this novel perspective of reflecting on representational similarity through the lens of functional similarity. A brief outline of this work[1] is the following:

- After a detailed introduction to model stitching (Section 3) and presenting two types of stitching methods (Section 4), we empirically demonstrate that trained convolutional networks with the same architecture but different initializations can be stitched together, even with a single, affine stitching layer in many cases without significant performance loss (Section 5). We will refer to this compatibility property as the 'matchability' of representations.

- Observing the matchability of representations as a quite robust property of common vision models, we have a wide range of examples in our hands when we know that representations are functionally similar. This permits us to study the relation between representational similarity measures and functional similarity. In our experiments, we show that the values of similarity indices are not necessarily indicative of task performance on a stitched network, and perhaps more surprisingly, high-performance stitchings can have differences in the values of similarity indices such as Centered Kernel Alignment (CKA) [Cortes et al., 2012, Kornblith et al., 2019]. This also reflects on phenomena experienced by the 'end-users' of these indices. For instance, in the context of continual learning Mirzadeh et al. [2021] observe that CKA remains constant, while the accuracy on previous tasks drops drastically (Section 6).

- Constraining the space of transformations in the stitching layer provides means to analyse how the information is organised in the representations. As an example of this probing methodology, we study bottleneck stitching layers, and show that when doing SVD in the representational space, the magnitudes of the principal components are not in direct correspondence with the information content of the latent directions (Section 7). As SVD is commonly used with embeddings, and there are representational similarity measures, for example SVCCA [Raghu et al., 2017], that use SVD as an ingredient, these results might be of interest to both theorists and practitioners.

- The transformation matrices of well-performing stitching layers capture how representations are functionally similar. This provides the opportunity to empirically investigate *"in what way are two representations similar, once we know that they are similar?"*. As a first step on this avenue, we present results regarding the uniqueness and sparsity properties of the stitching matrices (Section 8).

## 2   Related work

**Matching representations**   The idea of 1x1 convolution *stitching layers* connecting the lower half of a network with the upper half of another network first appears in Lenc and Vedaldi [2019]. They demonstrate that such hybrid "Franken-CNNs" can approach the accuracy of the original models in many cases. For shallower layers, this phenomenon can appear even when the first network is trained on a different task, suggesting that lower level representations are sufficiently generic.

*Mapping layers* as introduced by Li et al. [2016] are single layer linear 1x1 convolutional networks mapping between two representations. They do not incorporate information from the second half of the network. The authors train mapping layers with L1 regularized MSE, given their main goal of identifying sparse mappings between the filters of two different networks. As Li et al. [2016] notes, learning mapping layers can get stuck at poor local minima, even without L1 regularization. This forces the authors to restrict their experiments to the shallowest layers of the networks they inspect. We manage to overcome the issue of poor local optima by initializing from the optimal least squares matching (Section 8), and as our experiments demonstrate, this alleviates the issue, resulting in stitched networks that often achieve the performance of their constituents.

Concurrently and independently of our work, Bansal et al. [2021] also apply model stitching to investigate similarities between deep learning models. They observe that for a given task, even very different architectures can be successfully stitched, e.g., a transformer with a residual network.

---

[1]Code is available at the project website: `https://bit.ly/similarity-and-matching`.

**Similarity indices** Canonical Correlation Analysis (CCA) provides a statistical measure of similarity. CCA itself has a tendency to give too much emphasis to the behavior of low-magnitude high-noise latent directions. Similarity indices were designed to make CCA more robust in this sense, namely Singular Value CCA [Raghu et al., 2017] and Projection Weighted CCA [Morcos et al., 2018].

The current state of the art in measuring similarity is Centered Kernel Alignment (CKA) [Cristianini et al., 2001, Cortes et al., 2012, Kornblith et al., 2019]. Kornblith et al. [2019] conducted a comparative study of the measures CCA, SVCCA, PWCCA as similarity indices, and they observed that the other similarity indices do not succeed in identifying similar layers of identical networks trained from different initializations, whereas CKA manages to do so, therefore in this paper we chose to focus on the latter. Kornblith et al. [2019] also consider nonlinear CKA kernels, but as they demonstrate this does not lead to meaningful improvements, so we focus only on the linear variant (also commonly known as RV coefficient Robert and Escoufier [1976]), in line with other literature employing CKA similarity [Nguyen et al., 2021, Mirzadeh et al., 2021, Neyshabur et al., 2020, Wu et al., 2020].

**Functional similarity** The general framework of knowledge distillation [Hinton et al., 2015, Romero et al., 2014] can be seen as fostering end-to-end functional similarity of a student network to a teacher network. This relation and the versatile utilisation of this technique underpins the importance of the notion of functional similarity. Yosinski et al. [2014] studies transfer learning from a related perspective.

# 3 Introducing Dr. Frankenstein

## 3.1 Preliminaries and notation

Let $f_\theta : \mathcal{X} \to \mathcal{Y}$ denote the input-output function of a feedforward artificial neural network with $m \in \mathbb{N}$ layers, thus, $f_\theta = f_m \circ \cdots \circ f_1$, where $\mathcal{X}$ is the input space, $\mathcal{Y}$ is the output space, $f_i : \mathcal{A}_{i-1} \to \mathcal{A}_i$ are maps between activation spaces $\mathcal{A}_{i-1}$ and $\mathcal{A}_i$ for $i \in [m]$ with $\mathcal{A}_0 = \mathcal{X}$, and $\theta$ are the parameters of the network. We consider models that are trained on a dataset $D = \{(x_i, y_i)\}_{i=1}^n$ in a supervised manner with inputs $x_i \in \mathcal{X}$ and labels $y_i \in \mathcal{Y}$, where $n \in \mathbb{N}$ is the dataset size.

The central tool of our investigation involves splitting up the network at a given layer to a *representation map* and a *task map*. The *representation map at layer $L$* is a function $R_L : \mathcal{X} \to \mathcal{A}_L$ that maps each data point $x_i$ to its *activation vector* $a_{i,L}$ at layer $L$:

$$R_L(x_i) = f_L \circ \cdots \circ f_1(x_i) = a_{i,L},$$

where $\mathcal{A}_L$ is the activation space at layer $L$. The activation vectors for the dataset $D$ at layer $L$ are simply denoted by $A_L = (a_{i,L})_{i=1}^n$. The *task map at layer $L$* is a function $T_L : \mathcal{A}_L \to \mathcal{Y}$ that maps an activation vector at layer $L$ to the final output of the network:

$$T_L(a_{i,L}) = f_m \circ \cdots \circ f_{L+1}(a_{i,L}).$$

Thus, the input-output map of the network $f_\theta$ is simply the composition $f_\theta = T_L \circ R_L : \mathcal{X} \to \mathcal{Y}$ for all $L = 1, \ldots, m$. We will omit the index of the layer $L$ when this does not hurt clarity.

## 3.2 The Frankenstein learning task

**Definition 3.1** (Frankenstein network). *Fix two (pretrained) neural networks $f_\theta$ and $f_\phi$. We will call a transformation $S : \mathcal{A}_{\theta,L} \to \mathcal{A}_{\phi,M}$ between the two activation spaces at layers $L$ and $M$ a* stitching layer. *Given a stitching layer $S$, the Frankenstein or* stitched network *corresponding to $S$ is the composition $F = F_S : \mathcal{X} \to \mathcal{Y}$,*

$$F(x) = T_{\phi,M} \circ S \circ R_{\theta,L}(x),$$

*where $T_{\phi,M}$ is the task map at layer $M$ for the network $f_\phi$ and $R_{\theta,L}$ is the representation map at layer $L$ for network $f_\theta$.*

The stitching layer thus realizes a correspondence between the representations of two neural networks: it transforms the activations of one network at a particular layer to be a suitable input to the corresponding layer in the other network. For better readability, we will often call $f_\theta$ as Model 1 and $f_\phi$ as Model 2, and index the corresponding data accordingly ($f_1 = f_\theta$, $f_2 = f_\phi$, etc.). In the experiments of this paper, Model 1 and Model 2 have the same architecture and $L = M$.

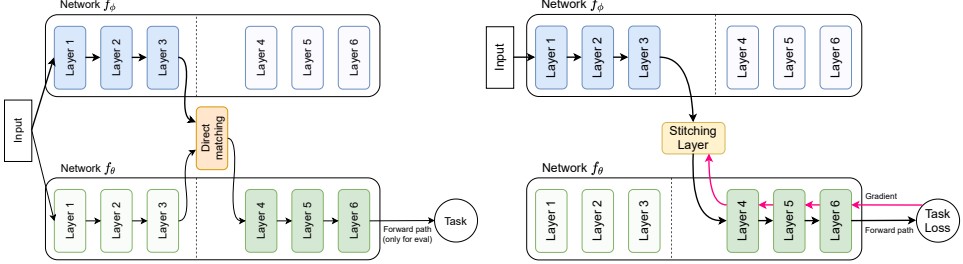

(a) Direct matching of representations.    (b) Task loss matching of representations.

Figure 1: Illustrating two types of matching approaches. **(a) Direct matching**: first align the representations using only the representations themselves, and use the forward path on the task map of Model 2 — with the transformed activations — only to evaluate the performance of the stitched model. **(b) Task loss matching**: the forward path from the input starts on Model 1, then through the stitching layer it continues on Model 2, then task loss gradients are backpropagated to update the weights of the stitching layer.

**Definition 3.2** (Frankenstein learning task). *Given two pretrained neural networks $f_\theta$ and $f_\phi$, a task given by a labeled dataset $D = \{(x_i, y_i) \in \mathcal{X} \times \mathcal{Y} : i = 1, \ldots, n\}$ and a loss function $\mathcal{L} : \mathcal{Y} \times \mathcal{Y} \to \mathbb{R}$, we will call the* Frankenstein learning task at layer $L$ *the task of finding the stitching layer $S : \mathcal{A}_{\theta,L} \to \mathcal{A}_{\phi,L}$ which minimizes the total value of the loss $\mathcal{L}$ on the dataset $D$.*

**Frankenstein is an evaluation framework**  The Frankenstein learning task primarily provides an *evaluation framework* for matching representations, and it does not tie our hands on how the transformation function $S$ is produced. As an evaluation metric, we can either use the mean loss value of the Frankenstein learning task on an evaluation set, or any other appropriate metric (e.g., accuracy in the case of classification tasks).

**Stitching convolutional representations**  Our toolset is generic, but here we will only consider convolutional architectures. Thus, the activation spaces have the structure of rank-3 tensors $\mathbb{R}^{w \times h \times c}$, where $w$ and $h$ are the spatial width and height and $c$ is the number of feature maps. We will only consider linear stitching layers of the form $M : \mathbb{R}^c \to \mathbb{R}^c$ ($1 \times 1$ convolutional layers). When formulating least squares problems on $n$ many activation tensors, we always mean the least squares problem in $\mathbb{R}^c$, which in practice means reshaping the $n \times w \times h \times c$ tensor to a $nwh \times c$ matrix.

## 4 Methods to solve the Frankenstein learning task

Here we line up several possibilities to solve the Frankenstein learning task presented in the previous section. We distinguish two approaches: **(1) Direct matching**: utilize only the outputs of representation maps; **(2) Task loss matching**: utilize the loss of the Frankenstein learning task. Figure 1 illustrates the difference between the two approaches schematically.

### 4.1 Task loss matching

From the exposition of the Frankenstein learning task, a training method naturally follows: utilize the loss function in the Frankenstein learning task itself, and train an appropriately parametrised stitching layer with backpropagation (leaving the other parts of the Frankenstein network fixed).

We use the outputs of Model 2 as a soft label [Hinton et al., 2015] and define the task loss with cross-entropy. Another meaningful option would be to use the original training objective (the one that was used for the training of Model 1 and Model 2, e.g., the one-hot labels in a classification task). Both are reasonable choices as tasks, one corresponding to the stitched network imitating Model 2, the other to solving the task itself. We have experimented with both losses and found very high correlation, with no difference regarding our main observations. We often present performance in terms of relative accuracy compared to Model 2, because it is easier to interpret than cross-entropy.

## 4.2 Direct matching methods

Due to their close connection to similarity indices of representations, we also consider methods that arrive at the transformation $S$ by using only the representations themselves, namely, utilizing only the outputs $A, B \in \mathbb{R}^{n \times p}$ of the representation maps of Model 1 and Model 2, respectively.

**Least squares methods**    Given $A, B \in \mathbb{R}^{n \times p}$, and a class of transformations $\mathcal{C} \subseteq \mathrm{Hom}(\mathbb{R}^p, \mathbb{R}^p)$, we will consider least squares problems of the following kind: find an optimal $M_o \in \mathcal{C}$, such that

$$\|AM_o - B\|_F = \min_{M \in \mathcal{C}} \|AM - B\|_F. \tag{1}$$

We consider three variants of this problem: 1. arbitrary linear maps: $\mathcal{C} = \mathrm{Hom}(\mathbb{R}^p, \mathbb{R}^p)$, 2. orthogonal transformations: $\mathcal{C} = O(p)$, 3. linear maps in $\mathrm{Hom}(\mathbb{R}^p, \mathbb{R}^p)$ of rank at most $k$: $\mathcal{C} = \Sigma_k$.

In each case, there is a closed form expression relying on singular value decomposition. For $\mathcal{C} = \mathrm{Hom}(\mathbb{R}^p, \mathbb{R}^p)$, the minimal value of (1) is obtained through an orthogonal projection in the space of matrices [Penrose, 1956],

$$M_o = M_{LS} := A^\dagger B, \tag{2}$$

where $A^\dagger$ denotes the Moore-Penrose pseudoinverse of $A$. We will refer to this $M_{LS}$ as the *least squares matching*. For results of this type, see Section 5.

In the case $\mathcal{C} = O(p)$, problem (1) is also known as the *orthogonal Procrustes problem*, see e.g. Golub and Van Loan [2013]. If $A^T B = USV^T$ is a singular value decomposition, then the optimal orthogonal map is $M_o = UV^T$.

Finally, the case $\mathcal{C} = \Sigma_k$ is also known as *reduced rank regression* Izenman [1975]. In terms of the SVD of $AM_{LS} = USV^T$, the optimal rank $k$ map $M_o$ is given by $M_o = M_{LS}V_k V_k^T$, where $V_k$ consists of the first $k$ columns of $V$. For reduced rank matching results, see Section 7.

**Sparse matchings**    It is known [Wang et al., 2018] that representations are not fully local, that is, matching cannot be achieved by permutation matrices. High quality sparse matchings between neurons can be interpreted to imply "slightly distributed" representations [Li et al., 2016].

To achieve sparsity, a natural approach is to add the regularization term $\|M\|_1 = \sum |m_{ij}|$ to the loss function, where $M$ is the transformation matrix. Then one can consider the L1 regularized versions of 1) task loss matching, and 2) least squares matching (Lasso regression, as in Li et al. [2016]):

$$\mathcal{L}(A, B) = \|AM - B\|_F + \alpha \cdot \|M\|_1, \tag{3}$$

where $A, B$ denote the activations of the two models. See Section 8 for these experiments.

## 5 Matching representations in deep convolutional networks

We now continue our exposition by presenting a series of experiments regarding matching neural representations. Our results in this section can be summarized in the following statement:

*Neural representations arising on a given layer of convolutional networks that share the same architecture but differ in initialization can be matched with a single affine stitching layer, achieving close to original performance on the stitched network.*

### 5.1 Experiment 1: Matching with least squares and task loss matching

In this experiment, we compare the performance of direct matching and task loss matching. We take two networks of identical architectures, but trained from different initializations. We perform a direct matching as well as a task loss matching from Model 1 to Model 2. We then compare the performance of the stitched network to the performance of Model 2.

We conduct experiments on three different convolutional architectures: a simple, 10-layer convnet called Tiny-10 (used in Kornblith et al. [2019]), on a ResNet-20 [He et al., 2016], and on the Inception V1 network [Szegedy et al., 2015]. These networks are rather standard and represent different types of architectures. Tiny-10 and ResNet-20 are trained and evaluated on CIFAR-10, Inception V1 on the 40-label CelebA task. Further training details follow common practices (see Appendix A.3).

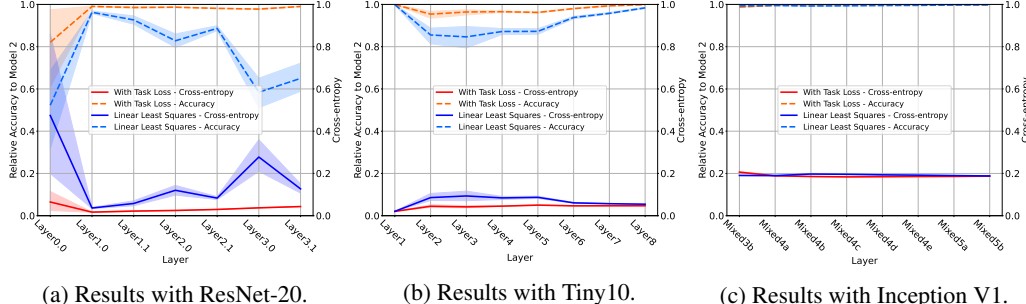

(a) Results with ResNet-20.  (b) Results with Tiny10.  (c) Results with Inception V1.

Figure 2: Performance of Frankenstein networks stitched together at specific layers (horizontal axis) measured in relative accuracy and cross-entropy (left and right vertical axes, respectively). This experiment compares two methods: task loss matching and linear least squares matching (a direct matching method). Results are averages of 10 runs, the bands represent standard deviations. As baseline values for cross-entropy, we provide average cross-entropies between Model 2 and a model stitched without transformation: Tiny-10: $6.99 \pm 1.96$, ResNet-20: $4.69 \pm 2.91$, InceptionV1: $0.36 \pm 0.08$ (averaged across all layers).

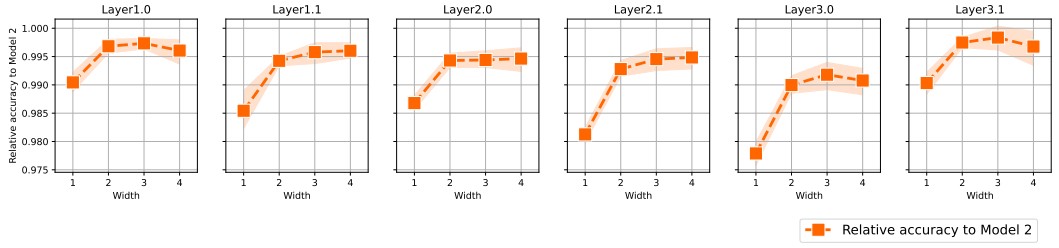

Figure 3: Relative accuracy values with ResNet-20 nets of different width. The horizontal axis shows the width multiplier compared to the baseline network. Plots correspond to different layers with names denoted at the top. Averages of 10 runs, bands show standard deviations.

Figure 2 summarizes the results: task loss matching achieves near perfect accuracy on ResNet-20, and also succeeds in aligning representations on Tiny-10 without collapsing accuracy. While the direct matching significantly underperforms the task loss matching and collapses performance on several layers, it is still remarkable that direct matching performs fairly well in many cases: e.g., on the layers between Layer1.0 and Layer2.1 on ResNet-20 or almost every case on Tiny-10.

**Further results**  We have experimented with networks with batch normalization and without batch normalization, and networks trained with different optimizers (SGD, Adam). Still, we did not encounter any cases where there were unmatchable instances. The worst performing stitchings were after the first convolutional layer on ResNet-20, still achieving 80% relative accuracy. While this method is not universal on arbitrary architectures, this does not modify the observation that 'matchability' of representations is a quite robust property of common vision models.

In the case of task loss matching, we found the most stable results when we initialized the training of the stitching layer from the least squares matching (2). See Section 8 on further details on how the initialization influences performance. We also evaluated Procrustes matching described in Section 4.2, and found that on later layers, it achieves the same level of accuracy as linear least squares matching.

As detailed in Section C.2 of the Appendix, accuracy drop during stitching can be surprisingly low even when we stitch networks trained on different datasets; in our case, ImageNet and CelebA.

## 5.2 Experiment 2: Networks of different width

This experiment focuses on the effect of network width on stitching performance. Matching happens with cross-entropy to Model 2 output as the task loss. The stitching is initialized to the optimal least squares matching. Performance is measured on the original classification task, relative to Model 2 performance. Figure 3 shows the results on ResNet-20 with various widths. The relative accuracy for 1x-width is already very high, but a clear trend of performance increasing with width is still visible.

# 6 Similarity indices and matching representations

We now turn our attention to the relation of similarity and matchability of representations. Finding an appropriate similarity index for neural representations is an active area of research [Gretton et al., 2005, Raghu et al., 2017, Morcos et al., 2018, Kornblith et al., 2019]. In the pursuit of a suitable notion, one seeks to capture the right invariance and equivariance properties to provide a tool with theoretical guarantees and practical applicability. Our brief summarizing statement in this section acts as a warning for the end-users of statistical similarity indices:

*By their nature, similarity indices depend on the task only indirectly, and their values are not necessarily indicative of task performance on a stitched network.*

## 6.1 Similarity indices

Given two sets of activations $A = (a_i)_{i=1}^n$, $B = (b_j)_{j=1}^n$, in the activation spaces $a_i \in \mathcal{A}_1 = \mathbb{R}^s$, $b_j \in \mathcal{A}_2 = \mathbb{R}^r$, a *similarity index* $s(A, B) \in [0, 1]$, measures the similarity of the two data representations, i.e. it is a statistic on the observation space $s : \mathcal{A}_1 \times \mathcal{A}_2 \to [0, 1]$. We now briefly review the similarity indices under consideration. First, the least squares matching $M_{LS}$ defined earlier can be used to define a statistic that can be used as a similarity index [Kornblith et al., 2019]:

**Definition 6.1.** The coefficient of determination *of the best fit is given in terms of the optimal least squares matching* $M_{LS} = A^\dagger B$ *as:*

$$R_{LR}^2(A, B) = 1 - \frac{\|AM_{LS} - B\|_F^2}{\|B\|_F^2}.$$

Let $k, l$ be positive definite kernels on a space $\mathcal{X}$ and let $K_{ij} = k(x_i, x_j)$ and $L_{ij} = l(x_i, x_j)$ be the Gram matrices of a finite sample $x_i \in \mathcal{X}$, $i = 1, \ldots, m$ drawn from a distribution on $\mathcal{X}$. The (empirical) *alignment of the Gram matrices* $K, L$ was defined by Cristianini et al. [2001] as

$$\mathrm{KA}(K, L) = \frac{\langle K, L \rangle_F}{\|K\|_F \cdot \|L\|_F}.$$

The importance of centering the kernels in practical applications was highlighted in Cortes et al. [2012]. Given a Gram matrix $K$, denote the *centered Gram matrix* by $K^c := CKC$, where $C = I_m - \frac{\mathbf{1} \cdot \mathbf{1}^T}{m}$ is the centering matrix of size $m$. The *Centered Kernel Alignment of the Gram matrices* $K, L$ is defined [Cortes et al., 2012] as $\mathrm{CKA}(K, L) = \mathrm{KA}(K^c, L^c)$. If the kernels are linear on their respective feature spaces, and if $X, Y$ are samples of the distribution in these feature spaces, then $\mathrm{CKA}(K_X, L_Y) = \mathrm{KA}(K_{XC}, L_{YC})$, using that $K_X^c = K_{XC}$. In particular, in the case of linear kernels CKA is also referred to as the *RV coefficient* [Robert and Escoufier, 1976]. Furthermore, if $X, Y$ are the *centered* matrices representing the samples of the distribution in the feature space, CKA can be given by the following definition:

**Definition 6.2.** *The* empirical (linear) Centered Kernel Alignment *or* RV coefficient *of the samples* $X, Y$ *centered in feature space is*

$$\mathrm{RV}(X, Y) := \mathrm{CKA}(K_X, K_Y) = \frac{\|X^T Y\|_F^2}{\|X^T X\|_F \cdot \|Y^T Y\|_F}.$$

In our context, the probability distribution describes the data distribution in $\mathcal{X}$, the feature spaces are the activation spaces and the feature maps are the representation maps $R : \mathcal{X} \to \mathcal{A}$.

As a measure of similarity of representations, CKA was introduced and used successfully in Kornblith et al. [2019]. They also considered other similarity indices such as CCA, SVCCA, PWCCA [Raghu et al., 2017, Morcos et al., 2018] in their experiments, but they found that the other similarity indices do not succeed in identifying similar layers of identical networks trained from different initializations, whereas CKA manages to do so. For this reason, we chose to focus our attention on CKA.

## 6.2 Experiment 3: Similarity indices and task performance

In this set of experiments, we monitor the CKA, $R_{LR}^2$, CCA and SVCCA indices throughout the training process of the stitching layer trained with task loss matching. We measure these values

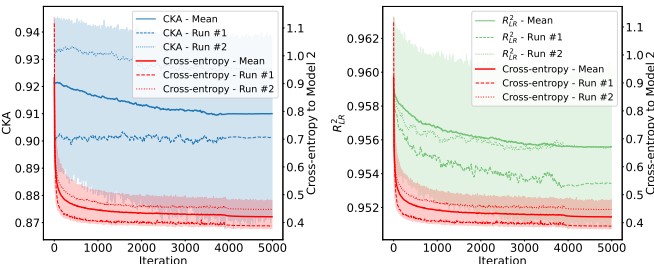
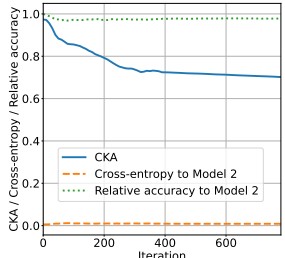

Figure 4: CKA, $R_{LR}^2$ and cross-entropy values over the training iterations of task loss matching started from the optimal least squares matching. Tiny-10 network, stitching at Layer 3. Bold lines are averages of 10 runs, with bands representing standard deviations; dotted and dashed lines correspond to two specific runs to show the individual characteristics of the training of the stitching layer. Further experiment details are in Appendix A.5.

Figure 5: CKA and performance measures throughout the training for a Frankenstein task with a loss that explicitly penalizes CKA value while maintaining the model performance. Stitching is at Layer 2.0 of a 3-wide ResNet.

between Model 2 activations and the transformed activations of Model 1. Figure 4 shows the results for CKA and $R_{LR}^2$. We observe that CKA has a large variance from the first step of the training process, and it is decreasing while the performance of the stitched model is increasing.

This experiment also highlights that $R_{LR}^2$ can correlate inversely with the performance: as the training proceeds with decreasing cross-entropy, the $R_{LR}^2$ values also decrease. Indeed, since we initialized the task loss training from the optimal least squares matching, the distances naturally must increase under any training process. However, this highlights the fact that the least squares matching is ignorant of the decision boundaries in a neural network.

CCA is invariant to invertible linear transformations, hence it ignores our linear stitching transformation during the training. SVCCA also performs CCA on truncated singular value decompositions of its inputs. Because its value only depends on the SVD step, it performs quite big jumps during the training, but on average it does not correlate with the task performance. See further results and experimental details in Appendix A.5.

### 6.3 Experiment 4: Low CKA value with high accuracy

We now show that it is easy to align representations in a way that results in high task performance while the CKA value at the alignment is small. The setup is the following: we take one trained model and stitch it with *itself*, utilizing a linear stitching layer initialised with the identity mapping. With this initialization we essentially reproduce the original model, thus, this trivially results in a Frankenstein network with the same performance as the original model. We train this stitching layer with an additional loss term that penalizes the CKA value at the stitching, meanwhile we maintain the task performance with the usual cross-entropy term. We use the CKA value itself as the penalty, as it is differentiable by definition and thus suitable for gradient-based optimization.

Figure 5 shows the outcome of such a training process. We find that the CKA value drops by a large amount (achieving lower than 0.7), whereas the accuracy remains essentially unchanged.

As a baseline, we provide the CKA values between representations of the transformed activations of Model 1 and Model 2 averaged over 10 different network pairs and all layers. ResNet: $0.922 \pm 0.098$, Tiny-10: $0.883 \pm 0.050$. For further baseline CKA values we also refer to Kornblith et al. [2019].

### 6.4 Discussion

Notions of representational similarity and functional similarity have different motivations, and thus, aim to capture different invariances. Part of these invariances could be compatible or incompatible. For example, assuming that a similarity index $s$ is invariant under a given class $\mathcal{C}$ of transformations, it is not surprising that the high value of a similarity index $s(A, B)$ does not indicate the relative performance of the activations $A$ and $B$: indeed, given a set of activations $A$, $s(AF, A) = 1$ for any $F \in \mathcal{C}$, while the task loss of the transformed activations $AF$ typically varies greatly. On the other hand, it is somewhat surprising that high-performance matchings can have a significant drop in the values of meaningful similarity indices such as CKA.

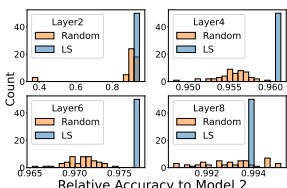

Figure 6: Relative accuracies by initialization type.

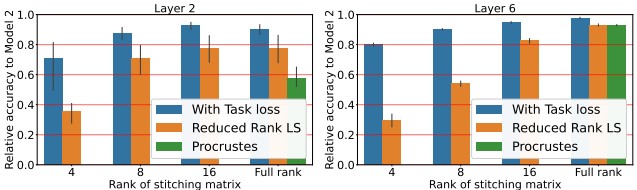

Figure 7: The performance of the low-rank analogues of least squares and task loss matchings in terms of relative accuracy.

**The inadequateness of CKA as a direct matching method** CKA is not directly suitable as an objective for direct matching, since it is invariant to orthogonal transformations. A stitching with an optimal CKA value can be arbitrarily transformed via orthogonal transformations without affecting CKA, but most of these equivalent solutions are unsuitable for the matching task as trained models are not invariant for arbitrary rotations in the activation spaces.

# 7 Low rank representations — a gap between SVD and task loss matching

In this experiment, we compare the effectiveness of the low-rank analogues of the above presented least squares matching and task loss matching. More precisely, we apply SVD directly in the activation space to obtain the low-rank least squares matching, and use a *bottleneck* stitching layer to realise a low-rank matching with the task loss.

Figure 7 depicts the results for two selected layers of Tiny-10 with prescribed ranks 4, 8, and 16. (We observed similar results for the other layers, and also for ResNet-20.) It is clear that task loss matching outperforms least squares matching in the activation space when the rank of the transformation is fixed. This suggests that the objective for the low-rank approximation of explaining most of the variance in the activation space (i.e., the objective of SVD) is not satisfactory to attain the highest possible performance with a prescribed rank. In other words, the magnitudes of the principal components are not in direct correspondence with the information content of the latent directions. Ultimately, this suggests that similarity indices that use SVD directly in the activation space — either as a preprocessing step or as another key element in the computation — might fail to capture some desired aspects of the representation. Details and figures for the other layers are in Appendix A.

# 8 Structure of stitching layers: uniqueness, mode connectivity, sparsity

The previous experiments show that the stitching layer can be trained to achieve the same performance as the original Model 2. What is the structure of the obtained stitching matrix? In this chapter, we explore the question of sensitivity with respect to initialization, uniqueness properties of the stitching layer and the relationship of direct and task loss matching with sparsity.

**Experiment 5: Dependence of accuracy on initializations** Are stitching layers obtained from different initializations the same? A crude measure of this is the performance of the stitching layer on the stitched network. In this experiment, we fix two identical network architectures trained from different initializations. We train the stitching layer on task loss from 50 random and 50 optimal least squares initializations, and compare their performance. We found that in most cases the stitched networks achieve the same order of accuracy, although not always; the training process might get stuck at poor local minima. However, the optimal least squares matching initialization performs consistently well. Figure 6 shows a histogram of the results. For further details, see Appendix A.

**Experiment 6: Linear mode connectivity of transformation matrices** We also investigated the uniqueness properties of the stitching layer by examining the linear mode connectivity of the found transformations. In particular, on a given layer of two trained copies of Tiny-10, we trained stitching transformations from different random initializations. We only kept the transformations achieving a relative accuracy of at least 80% — there is no reason to expect mode connectivity properties where there is a large difference in relative accuracy. We then evaluated the relative accuracy of the stitched network with the transformation matrix $M_\lambda = \lambda M_1 + (1 - \lambda) M_2$ for pairs of transformation matrices $M_1$ and $M_2$, and different values of $\lambda \in [0, 1]$. The results showed that mode connectivity

 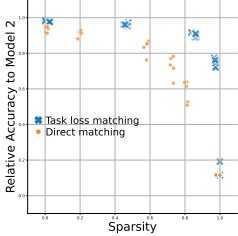

Figure 8: Comparison of stitching matrices and the relative accuracy of L1-regularized direct and task loss matching. To achieve sparsity, we set every entry of the stitching matrix to zero when the absolute value of the entry is below the threshold $10^{-4}$. Sparsity is the ratio between the zero elements and the total number of elements in the stitching matrix. Relative accuracies with zero sparsity correspond to the unregularised baseline. **Left:** Examples of stitching matrices with different sparsity. **Right:** The degradation of relative accuracy with respect to increasing sparsity is significantly less steep with task loss training. For more results, see Appendix A.9.

holds on most layers, i.e. the convex combination of the stitching layers perform similarly. However, on the first three layers, we noticed an unforeseen lack of mode connectivity, even though the relative accuracies of the corresponding transformations were high. For further details, see Appendix A.8.

**Experiment 7: Sparsity — direct vs. task loss matching** How does the sparsity of the stitching layer influence the performance of the stitched network? In this experiment, we achieve sparsity by adding an L1-regularizing term to the training as detailed in Section 4.2, similarly to the experiments in Li et al. [2016]. We perform both an L1-regularized task loss matching as well as a direct loss matching with loss term (3). To obtain a truly sparse stitching matrix we set the elements to zero below a threshold. Sparsity is then computed as the ratio between the zero elements and the total number of elements in the matrix. Figure 8 shows the dependence of relative accuracy on sparsity compared for task loss and direct matching as well as a sample of the stitching matrices. As Figure 8 shows, at around 80-90 % sparsity, the relative accuracy of direct matching suffers a drastic drop, however task loss matching still manages to find a meaningful alignment of the neurons using additional information via the task loss. These experiments show that even though it is possible to achieve a certain degree of sparse alignment of the neurons, the relationship between two representations is typically not of a combinatorial nature, and much can be gained by considering more complex relationships between the two layers. Furthermore, the difference in the relative accuracy of direct and task loss matching is notable for higher values of sparsity. For further details, see Appendix A.9.

## 9 Limitations

The performance of the stitched solution is inherently just a lower bound due to possible local minima. The current study is restricted to well-established convolutional architectures. Our observations on network width is analysed only in isolation.

## 10 Conclusions

We studied similarity of neural network representations in conjunction with the notion of matchability. We demonstrated the matchability of representations in convolutional networks through several experiments and analysed the properties of the stitching transformations with respect to the task performance. Regarding similarity notions, we provided a novel perspective which incorporates the performance on a task; a perspective which is yet unexplored in the field. We pointed out weaknesses of popular similarity notions that end-users might encounter.

## Broader impact

Our research is of foundational nature and advances machine learning at a fairly general level. While the authors do not foresee that the work herein will have immediate adverse ethical or societal consequences, such analysis of representation learning could help a malicious user avoid detection that is based on representational similarity; on the other hand, it could move forward the field towards more transparent and interpretable machine learning models.

## Acknowledgements

This work was supported by the European Union, co-financed by the European Social Fund (EFOP-3.6.3-VEKOP-16-2017-00002), the Hungarian National Excellence Grant 2018-1.2.1-NKP-00008 and by the Hungarian Ministry of Innovation and Technology NRDI Office within the framework of the Artificial Intelligence National Laboratory Program. We thank the anonymous reviewers for their many valuable suggestions.

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
