# A  Experiment details

## A.1  Datasets, splits, preprocessing, data augmentation

**CIFAR-10**  The CIFAR-10 dataset Krizhevsky [2012] consists of 60000 32x32 colour images in 10 classes, with 6000 images per class. There are 50000 training images and 10000 test images. We used the canonical train–test split. As a preprocessing, we normalized the images with the means (0.4914, 0.4822, 0.4465) and standard deviations (0.2023, 0.1994, 0.2010) for the three RGB channels, respectively. As augmentation, we used random horizontal flip, and 32x32 sized random crop from the zero padded 40x40 inputs.

**CelebA**  The CelebA dataset Liu et al. [2015] is a large-scale face attributes dataset, it consists of 202599 number of images depicting faces of celebrities, each with 40 attribute annotations. We used a random split with 80% train and 20% test set sizes. Originally the CelebA images are sized $178 \times 218$ (width $\times$ height). As a preprocessing, we first reshaped these images to 256x256 pixels and then applied a center crop to 224x224 which was our final input shape for the Inception V1 networks. We subtracted 117 from each pixel. As augmentation, we applied a random rotation between degrees of -10 and 10.

## A.2  Network architectures and training details

**Tiny-10**  Tiny-10 is a simple multi-layer convnet architecture. We use this model to provide a simple convnet for the experiments that trains fast on modern hardware. (A similar architecture was used also in Kornblith et al. [2019].) Table 1 details the layers along with the names we used in the paper to refer to a particular part of the network. While this naming is ambiguous as it could refer to three different activation spaces (in a row of the table), we use the activations after the batch normalization and before the nonlinearity if not otherwise stated.

We trained the model on CIFAR-10 for 300 epochs, the optimizer was SGD with Nesterov momentum 0.9. There was a schedule for the learning rate: started with the value of 0.1 and it was divided by 10 at 1/3 of the training, and with another 10 at the 2/3 of the training. The batch size was 128. We used weight decay with value $10^{-4}$. The average accuracy of the resulting models was 86.55%.

Table 1: The Tiny10 architecture.

| Tiny10 | |
|---|---|
| Layers | Name |
| $3 \times 3$ conv. 16-BN-ReLu | Layer 1 |
| $3 \times 3$ conv. 16-BN-ReLu | Layer 2 |
| $3 \times 3$ conv. 32 stride 2-BN-ReLu | Layer 3 |
| $3 \times 3$ conv. 32-BN-ReLu | Layer 4 |
| $3 \times 3$ conv. 32-BN-ReLu | Layer 5 |
| $3 \times 3$ conv. 64 stride 2-BN-ReLu | Layer 6 |
| $3 \times 3$ conv. 64 BN-ReLu | Layer 7 |
| $1 \times 1$ conv. 64-BN-ReLu | Layer 8 |
| Global average pooling | |
| Dense | |
| Logits | |

**ResNet-20**  Our ResNet-20 He et al. [2016] variant follows common practices regarding CIFAR-10: we use a 3-level architecture with three residual blocks per level. A residual block contains the following layers: Conv-Batchnorm-ReLU-Conv-Batchnorm. After each residual block, a ReLU operation follows the addition operation. Convolution kernels are sized 3x3. In the paper, we use the following naming convention: `LayerX.Y` corresponds to activations after the addition operation following a residual block with index `Y` in level `X`, with the exception of `Layer0.0`, which corresponds to the activation space after the first Conv-Batchnorm layer in the network preceding the residual blocks.

We trained the model on CIFAR-10 for 300 epochs, the optimizer was SGD with Nesterov momentum 0.9. There was a schedule for the learning rate: started with the value of 0.1 and it was divided by 10 at 1/3 of the training, and with another 10 at the 2/3 of the training. The batch size was 128. We used weight decay with value $10^{-4}$. The average accuracies were 91.95%, 93.97%, 94.59%, 94.77% for the 1-width, 2-width, 3-width, and 4-width ResNets, respectively.

**Inception V1**   For the Inception V1 Szegedy et al. [2015], we omit the detailed description of the architecture as a reiteration would be cumbersome, and there are no specifics for the task at hand. Regarding the layer names, we follow the standard naming conventions.

We trained the model on CelebA for 20 epochs, using the Adam optimizer with parameters $\beta_1 = 0.9$ and $\beta_2 = 0.999$. Learning rate was 0.0001, batch size was 128.

## A.3   Experiment 1 - Details for matching with least squares and task loss matching

In this experiment, we take network pairs (Model 1 and Model 2) which are of the same architecture but trained from different weight initializations and with different orderings of the training set.

**Least squares matching**   Let $A$ and $B$ denote the activation matrices for the training data of Model 1 and Model 2, respectively. We appended an all ones vector to the activation matrix $A$ of Model 1 to represent the bias. Then we calculated the pseudoinverse $A^\dagger$ of $A$ using SVD. The transformation matrix and the bias of the stitching layer was obtained by calculating $A^\dagger B$.

**Task loss matching**   We initialized the transformation matrix and the bias of the stitching layer to the least squares solution (which was calculated as described above). Then we trained the stitching layer on the train set for 30 epochs. The utilized loss was cross-entropy to Model 2 activations. The optimizer was Adam with parameters $\beta_1 = 0.9$ and $\beta_2 = 0.999$, learning rate was set to $10^{-3}$, batch size was 128.

We used the following hyperparameter selection protocol: after a grid search with Tiny-10 and ResNet consisting of the parameter settings { Optimizer: Adam, SGD } × { Learning rate: 0.1, 0.01, 0.001, 0.0001, 0.00001 } and training for 300 epochs, we observed that Adam is significantly better than SGD, and that learning rates below 0.001 do not affect performance, only prolong the training time. Moreover, we observed that with the selected hyperparameters the training of the stitching layer always reaches a plateau before the 30th epoch, thus, we set this hyperparameter accordingly.

On Tiny-10, we matched the activations after the batch normalization layer (which comes after the convolution and before the nonlinearity). With ResNets, we plotted the results that correspond to stitchings after the addition operations of the residual blocks. (Matchings on the inside layers of residual blocks are harder to interpret, while the results are very similar).

## A.4   Experiment 2 - Details for networks of different width

We utilized the same methodology and settings to train the stitching layer as described in Appendix A.3. The baseline 3-level ResNet had 16, 32, and 64 filters in the convolution layers for each level, respectively. The networks of different width were obtained by multiplying these baseline filter numbers with the width multiplier.

## A.5   Experiment 3 - Details for similarity indices and task performance

We utilized the same methodology and settings to set or train the stitching layer as described in Appendix A.3.

CKA, CCA and SVCCA is calculated on the whole validation set. We can fit this amount of data into memory without needing to resort to sampling methods like "minibatch CKA" Nguyen et al. [2021]. We note that the resulting CKA values are indistinguishable from values obtained when working with any number of data points between 2500 and 10000.

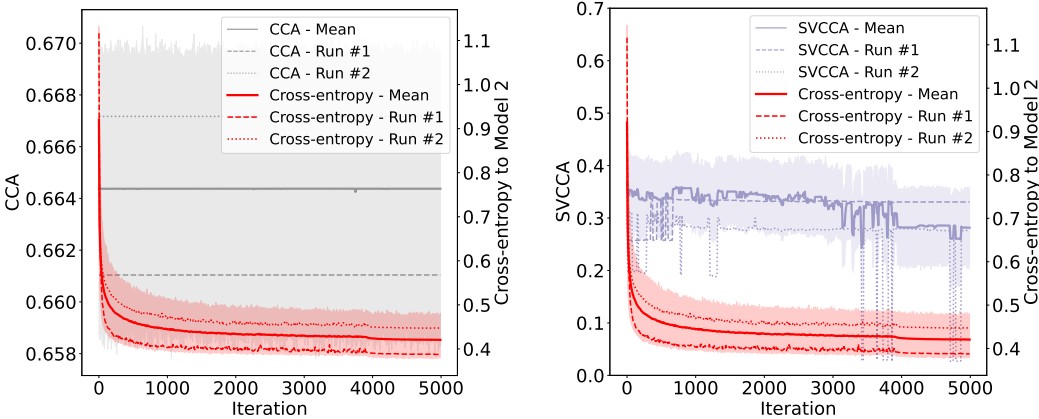

Figure 9: CCA, SVCCA and cross-entropy values over the training iterations of task loss matching started from the optimal Least Squares matching. Tiny-10 network, stitching at Layer 3. Bold lines are averages of 10 runs, with bands representing standard deviations; dotted and dashed lines correspond to two specific runs to show the individual characteristics of the training of the stitching layer.

### A.6 Experiment 4 - Details for low CKA value with high accuracy

The loss penalizing CKA was calculated on the training minibatch, the reported values were calculated on the whole validation set. We trained the stitching layer for 2 epochs with batch size 128, CKA loss weight 0.1, the optimizer was Adam with parameters $\beta_1 = 0.9$ and $\beta_2 = 0.999$, learning rate was 0.001 at the first epoch and 0.0001 in the second epoch. (Nevertheless, we observed that the general outcome of the experiment is quite robust for a large range of hyperparameters.)

### A.7 Experiment 5: Details for dependence of accuracy on initializations

We train the stitching layer on task loss from 50 random and 50 optimal least squares initializations between two Tiny-10 architectures trained from different initializations. Other settings of the experiment are the same as presented in Appendix A.3. Figure 10 shows the results for all layers, between three pairs of networks.

### A.8 Experiment 6: Details for linear mode connectivity of transformation matrices

We utilized the same methodology and settings to train the stitching layers as described in Appendix A.3. See Figure 11 for detailed results.

### A.9 Experiment 7: Details for sparsity — direct vs. task loss matching

To compare the sparsity tolerance of task loss matching and direct matching, we trained 5 stitching layers with both methods until convergence, which means 30 and 200 epochs, respectively. We used the Tiny-10 architecture, and the CIFAR-10 dataset. For task loss matching we utilized the same methodology and settings as described in Appendix A.3. We trained the direct matching using the Adam optimizer with parameters $\beta_1 = 0.9$ and $\beta_2 = 0.999$, learning rate was set to $10^{-2}$. We ran the trainings with different L1-regularization coefficients, namely: $0, 10^{-4}, 10^{-3}, 10^{-2}, 10^{-1}, 10^0,$ $10^1, 10^2, 10^3, 10^4$ to achieve increasing sparsity. In order to achieve an actual sparse matrix, we set every entry of the matrix to 0 below a certain threshold, $10^{-4}$. We evaluated the performance of the sparse stitching matrices on the task. Figure 12 shows the relative accuracy and sparsity (ratio of zero elements) with respect to L1-regularization term, and Figure 13 shows the relative accuracy with respect to sparsity for all matched layers.

## A.10 Experiment 8: Details for low rank representations

For task loss matching, the low rank transformation is ensured by a bottleneck in the stitching layer: for a prescribed rank $k \in \mathbb{N}$, an $n \times n$ transformation matrix is parametrized by the product of two $k \times n$ and $n \times k$ sized matrices.

For least squares matching, we use the reduced rank regression outlined in the main text.

In all other respects, we utilize the same methodology and settings to set or train the stitching layer as described in Appendix A.3.

Figure 14 depicts further results for all the layers of the Tiny-10 architecture for this experiment.

## A.11 Compute resources

We trained and evaluated approximately 20000 stitching layers overall. We used an internal cluster with GeForce 1080 Ti and GeForce 2080 Ti GPUs and dual Intel Xeon E5-2650 v4 CPUs in the machines. Each experimental run for a stitching layer with training and evaluation together used one (or a partial) GPU and generally finished under ten minutes for the Tiny-10 architecture, and under fifteen minutes for a 1-wide ResNet on these machines.

# B  Further direct matching methods

## B.1  Weighted mean squares matching of activations

In these experiments, we train the stitching layer to minimize the Weighted Mean Squared (WMS) objective. Our goal was to find a more or less simple method which sorts the activations, and assigns higher weights to the activations where a more precise matching is beneficial from the perspective of task performance. Roughly speaking, this method tries to identify more important features and put them into focus during the direct matching.

Given flattened activations $A, B \in \mathbb{R}^{n \times s}$ and a matrix of weights $W \in \mathbb{R}^{n \times s}$ we used Stochastic Gradient Descent to find the $M$ which minimizes the WMS distance between $AM$ and $B$:

$$\min_{M \in \mathbb{R}^{s \times s}} \|(AM - B) \circ W\|_F, \tag{4}$$

where $\circ$ is the Hadamard product $[X \circ Y]_{ij} = X_{ij} \cdot Y_{ij}$.

For the experiments listed below, we used the Tiny-10 architecture, the Adam optimizer with parameters $\beta_1 = 0.9$ and $\beta_2 = 0.999$, learning rate was set to $10^{-2}$, batch size was 64 and we trained for 200 epochs.

In the following, we will present different choices for the unnormalized weight matrices $W_u$. In each case, we will normalize the weight matrix with entries in $[0, 1]$ as follows:

$$W = \frac{W_u - w_{\min} \mathbf{1_N} \mathbf{1_s^T}}{w_{\max}}, \tag{5}$$

where $w_{min}, w_{max}$ denote the smallest and largest entries in $W$.

**Gradient based weighting**  As the Class Saliency method points out Simonyan et al. [2014], the gradients of a network's outputs with respect to its inputs may contain valuable information about the importance of each part of the input. Our method is analogous, however we inspect the hidden activations instead of the inputs. In this experiment, we solved a weighted mean squares matching (4) where the weights were determined from the gradients in the network.

In the following, fix the network weights $\phi$ and the layer $L$, and denote by $T_k = T^k_{\phi, L}$ the task map from $\mathcal{A}_{\phi, L}$ to the $k$th class, and $\hat{T}_k = \hat{T}^k_{\phi, L}$ the task map from $\mathcal{A}_{\phi, L}$ to the $k$th class without the last non-linearity. Denote by $\partial_{ij} h = \partial_j h(x_i)$ the partial derivative of a real-valued function $h : \mathcal{A}_{\phi, L} \to \mathbb{R}$ on the $i$th datapoint according to the $j$th coordinate in the activation space $\mathcal{A}_{\phi, L}$.

We experimented with four variants of gradient based weighting; in each case, we took as unnormalized weights $W^u$ a matrix of the form

$$W^u_{ij} = \partial_{ij} f(g_k),$$

where $f \in \{\sum_k, \text{argmax}_k\}$ and $g_k \in \{T_k, \hat{T}_k\}$. In each case we solved the weighted mean squares matching problem (4) by taking as weight matrix $W$ the normalization of such a weight matrix $W^u$. In particular, we considered the following variants:

- Gradients of summed output of Model 2 with respect to its matched activations:

$$W^u_{ij} = \partial_{ij} \sum_k T_k$$

- Gradients of the Model 2's output of the predicted class with respect to its matched activations:

$$W^u_{ij} = \partial_{ij} \text{argmax}_k T_k$$

- Gradients of summed output logits of Model 2 with respect to its matched activations:

$$W^u_{ij} = \partial_{ij} \sum_k \hat{T}_k$$

- Gradients of Model 2's output logits of the predicted class with respect to its matched activations:

$$W^u_{ij} = \partial_{ij} \text{argmax}_k \hat{T}_k$$

However, we found that overall these methods did not result in a consistent performance gain compared to the direct matching with unweighted least squares objective.

**Activation based weighting**    Another approach is to take as unnormalized weights $W_u = B$, and use the corresponding normalized weight matrix $W$ defined by (5). A simple way to emphasize higher values in this weight matrix is to take higher powers of each entry of $W$, i.e. use the weighting

$$W^{\circ n} = W \circ \ldots \circ W, \tag{6}$$

where $\circ$ denotes the (element-wise) Hadamard product.

We highlight some of these results for different values of $n$ in Figure 15a. Our experiments show, that matching Tiny-10 architecture's layers this way results in significant performance gain compared to the Least Squares based direct matching. This indicates that matching the higher regime of activations accurately is more important than the lower regime.

Another way to force higher activation focused matching is simply to define a threshold $T$, and only match the activations above this threshold. In particular, this corresponds to a $0 - 1$ weighting matrix

$$W = [1_{>T}(b_{ij})] \tag{7}$$

where $1_{>T}(x)$ denotes the indicator function, which serves as a threshold function. See Figure 15b for experiments with different thresholds $T_i$, defined by different percentiles $i \in \{10, 20\}$ of the activation values $b_{ij}$. We also tested higher threshold values defined by higher percentiles, which resulted in weaker performance. With this approach the activations below the threshold are not matched at all.

We also experimented with a slightly modified setup, where we define the weights as:

$$W_{ij} = 1 - 1_{<T}(b_{ij}) \cdot 1_{<T}([AM]_{ij}) = \begin{cases} 0, & [AM]_{ij} < T \text{ and } B_{ij} < T, \\ 1, & \text{else.} \end{cases} \tag{8}$$

This choice of weights achieves that if a target activation in $B$ is above a certain threshold, the corresponding activation distance $(b_{ij} - [AM]_{ij})^2$ is always matched with weight 1. However, if the target activation in $B$ is below the chosen threshold, then the direct matching only penalizes the corresponding distance if the matched activation $[AM]_{ij}$ exceeds the threshold. Informally, we don't care about the residuals as long as both activations are below the threshold. See Figure 15c for experiments with different thresholds $T_i$ defined by different percentiles $i \in \{10, 20, 30, 40, 50\}$. The higher thresholds resulted in weaker performance.

Another version of this last experiment (with weights (8)), is to set the threshold $T = 0$, which is a reasonable choice because Tiny-10 uses ReLU activations. See Figure 15d for results.

# C Further experiments

## C.1 A sanity check for stitching

We measured the accuracy of stitching with task loss between $N_i^1$ and $N_j^2$, where $N^1, N^2$ are networks with the same architecture, trained on the same data but with different sample order, and $N_i, N_j$ refers to the $i$th and $j$th layers of the network.

Stitched activations are required to have the same dimensionality, so we downsampled the higher dimensions with maxpooling before stitching. As upsampling without additional information is questionable, we always stitched from higher dimension (earlier layer) to lower dimension (later layer). Figure 16 shows the results measured on Resnet-20 architecture with width 1. We utilized the same methodology and settings to train the stitching layer as described in Appendix A.3.

## C.2 Cross-task stitching, Feature visualization

The following experiments use the Lucent port[2] of the Lucid interpretability framework [Olah et al., 2017] to visualize how Model 2 channels are constructed from the linear combinations of Model 1 channels during direct matching and stitching.

In the first setup, two Inception V1 networks trained on CelebA are stitched at various layers. The feature visualizations are shown in Figure 17. In the second setup, Model 1 is trained on ImageNet, and Model 2 is trained on CelebA. The feature visualizations are shown in Figure 18.

As seen in Tables 2a and 2b, the relative accuracies are consistently high, close to 100% in the case of the CelebA-CelebA stitchings. Even more notable is the case of the ImageNet-to-CelebA stitchings, where even the highest layers of the randomly initialized CelebA network can reuse the ImageNet features, with only a 3% relative drop in accuracy.

Note that a limitation of backpropagation-based feature visualization is that even though the visualized feature is a pattern that activates the channel, it can be activated by very different patterns as well.

| layer | Least Squares | Frankenstein |
|-------|---------------|--------------|
| 3a | 99.45% | 99.88% |
| 3b | 98.73% | 99.86% |
| 4a | 99.41% | 99.85% |
| 4b | 99.21% | 99.84% |
| 4c | 99.05% | 99.87% |
| 4d | 98.80% | 99.91% |
| 4e | 98.88% | 100.01% |
| 5a | 99.98% | 100.07% |
| 5b | 99.92% | 100.08% |

(a) Stitchings between two CelebA networks with different random initializations.

| layer | Least Squares | Frankenstein |
|-------|---------------|--------------|
| 3a | 97.72% | 99.77% |
| 3b | 95.90% | 99.62% |
| 4a | 97.73% | 99.41% |
| 4b | 96.74% | 99.22% |
| 4c | 95.27% | 98.88% |
| 4d | 93.45% | 98.42% |
| 4e | 94.37% | 98.21% |
| 5a | 95.75% | 97.49% |
| 5b | 95.57% | 97.02% |

(b) Imagenet to CelebA stitchings.

Table 2: Inception V1 stitchings, relative accuracies.

---

[2] https://github.com/greentfrapp/lucent

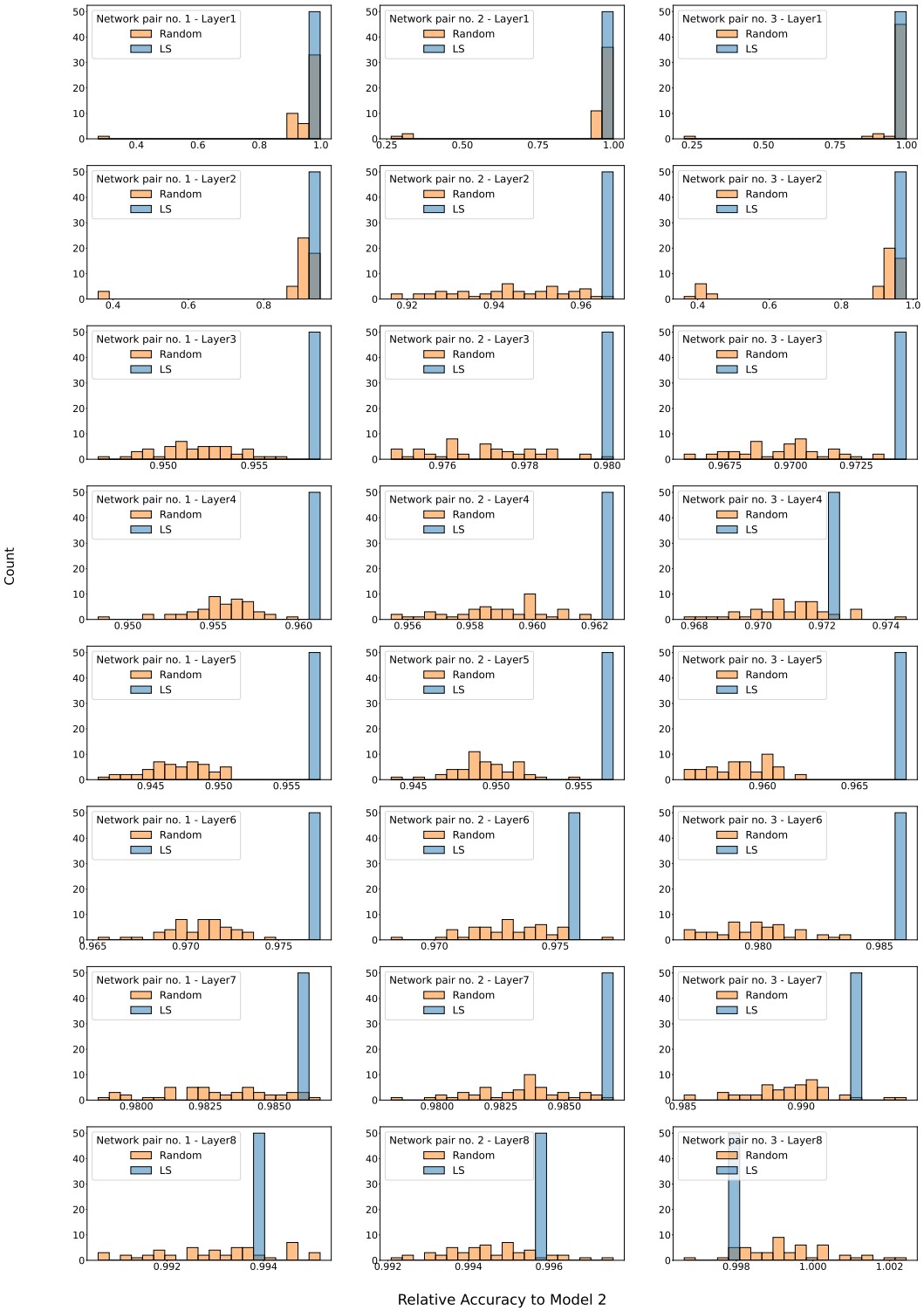

Figure 10: Comparing matching performance of randomly initialized transformation matrices versus transformation matrices initialized with the least squares solution. Different plots correspond to matchings on different layers, and different network pairs.

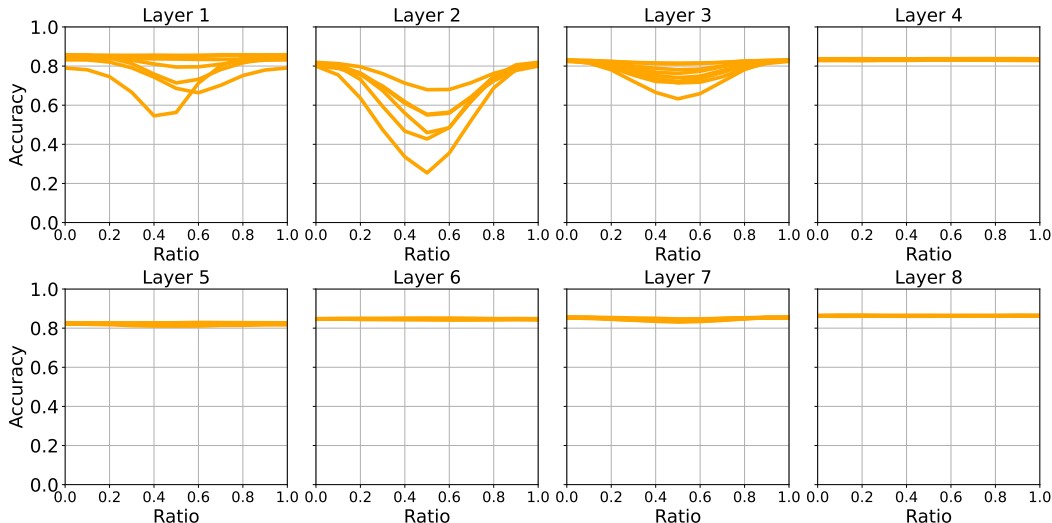

(a) Linear mode connectivity of stitched networks trained from different random initializations.

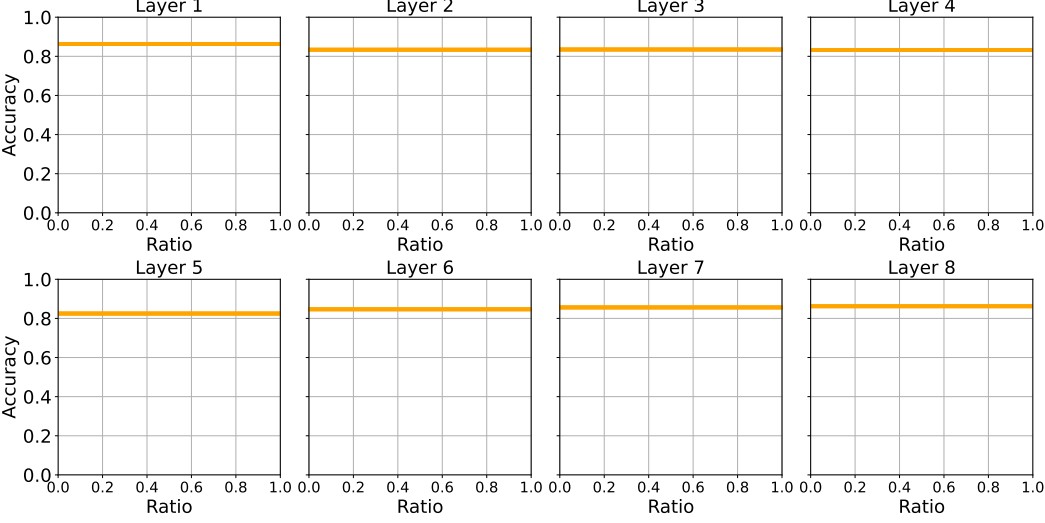

(b) Linear mode connectivity of stitched networks initialized with the least squares solution. Here the result is still not deterministic, because of the dataset iteration order randomness, which differs in each solution.

Figure 11: Relative accuracy of the stitched network with respect to $\lambda$, where the transformation matrix is $M_\lambda = \lambda M_1 + (1 - \lambda)M_2$ for pairs of transformation matrices $M_1$ and $M_2$. Different plots correspond to matchings on different layers.

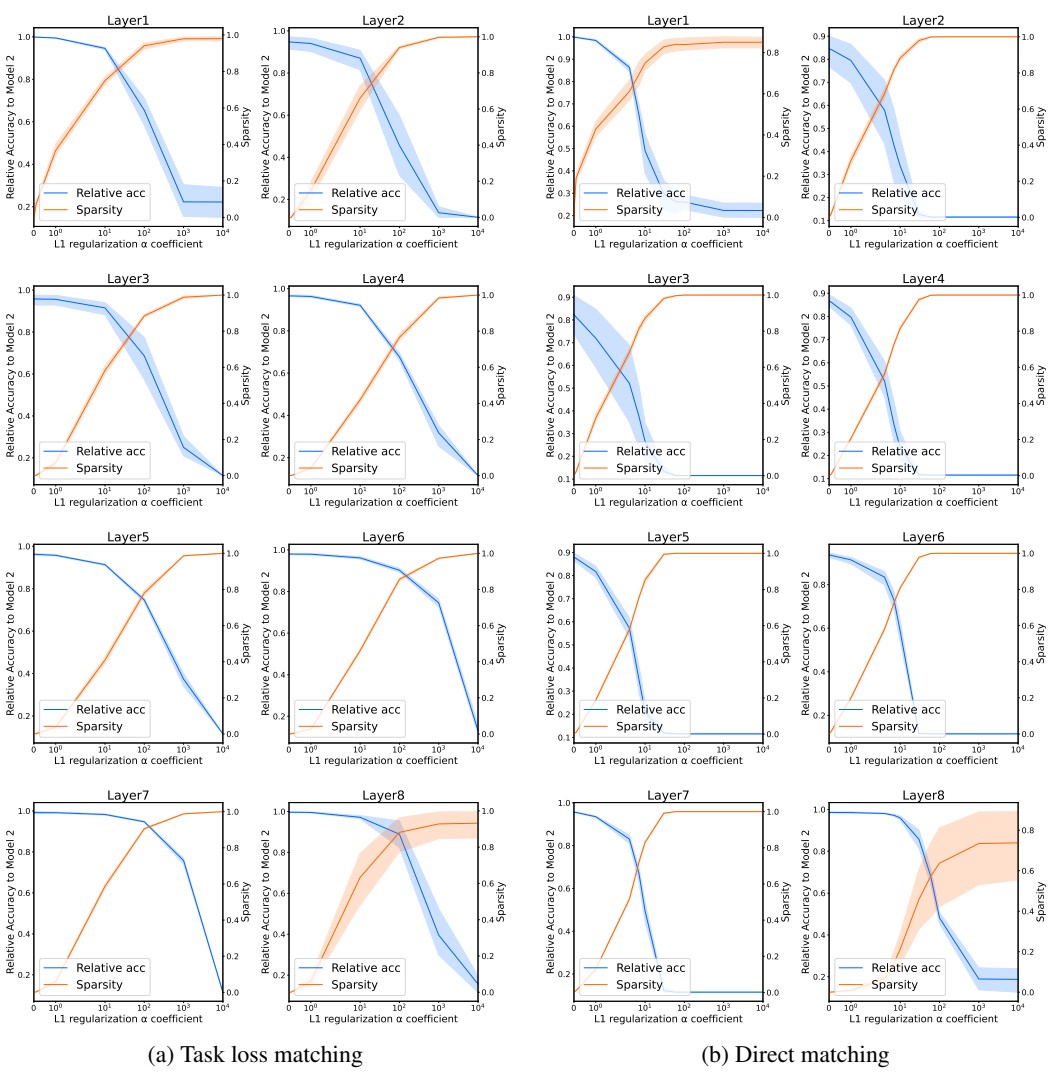

(a) Task loss matching          (b) Direct matching

Figure 12: Plotting relative accuracy to Model 2 and sparsity for different L1-regularization $\alpha$ coefficients. Different plots correspond to matchings on different layers of the Tiny-10 architecture. Results are averages of 5 runs, bands denote standard deviations.

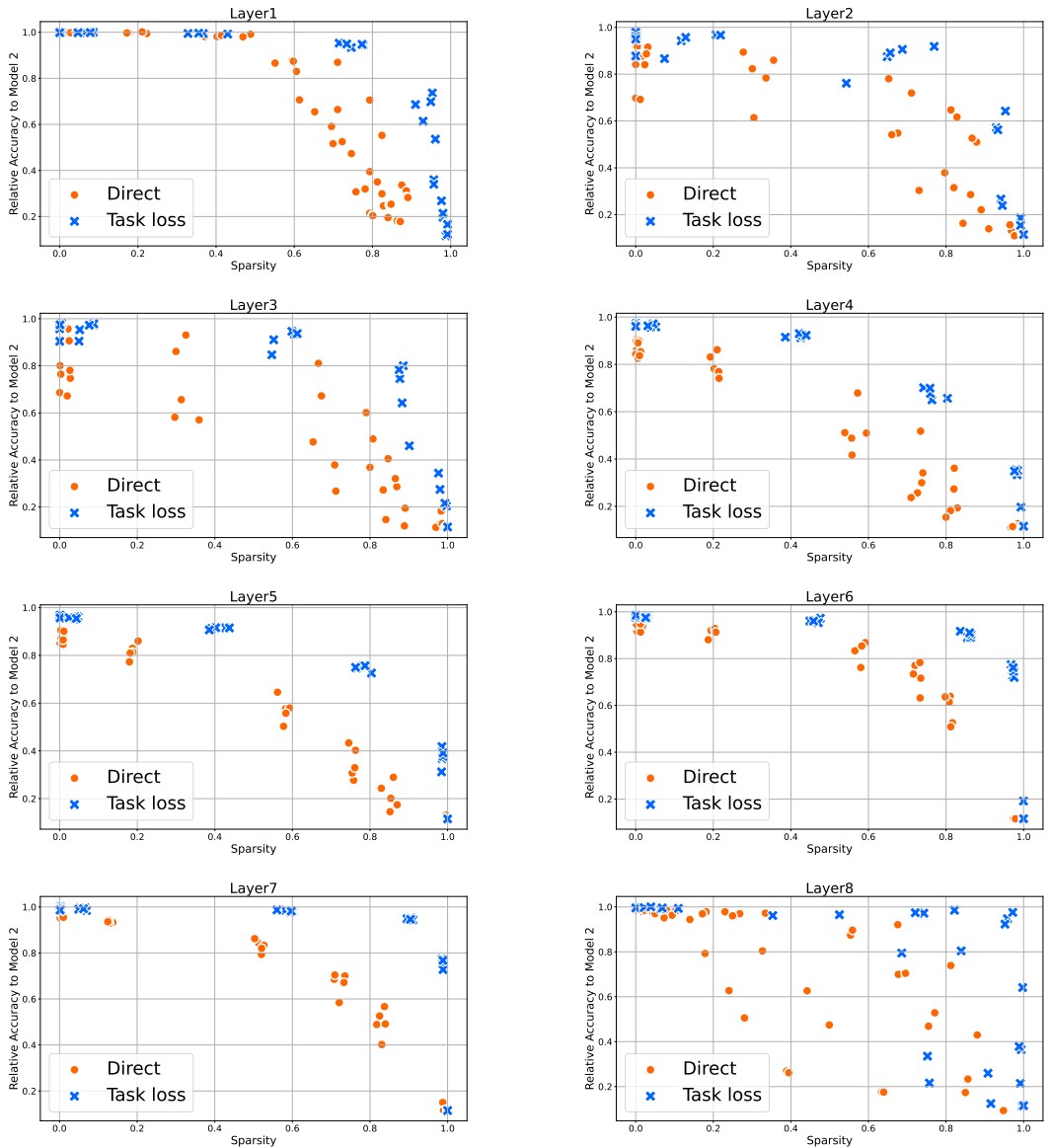

Figure 13: Relative accuracy to Model 2 with respect to sparsity. Different plots correspond to matchings on different layers.

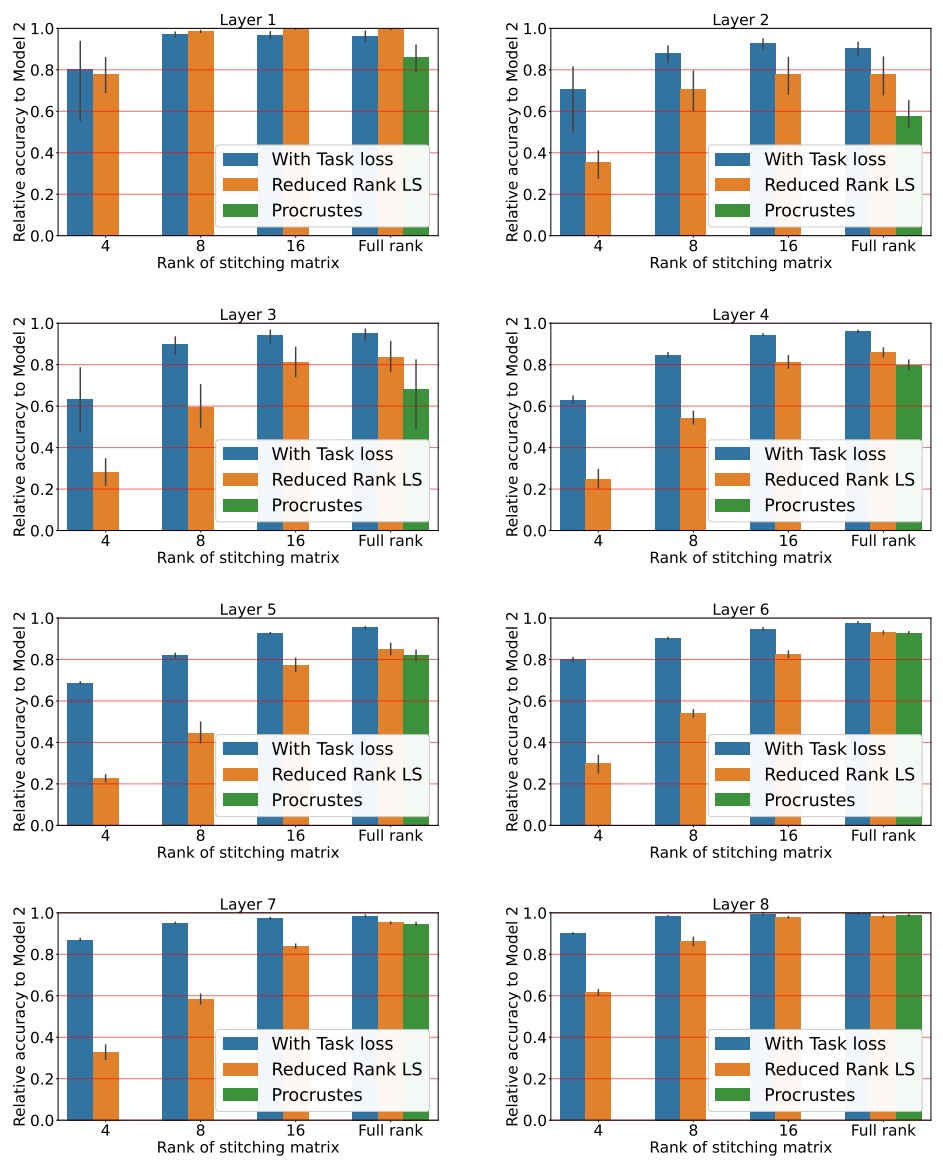

Figure 14: Performance of the low-rank analogues of least squares and task loss matchings in terms of relative accuracy. Averages of 5 runs, error bars denote standard deviations.

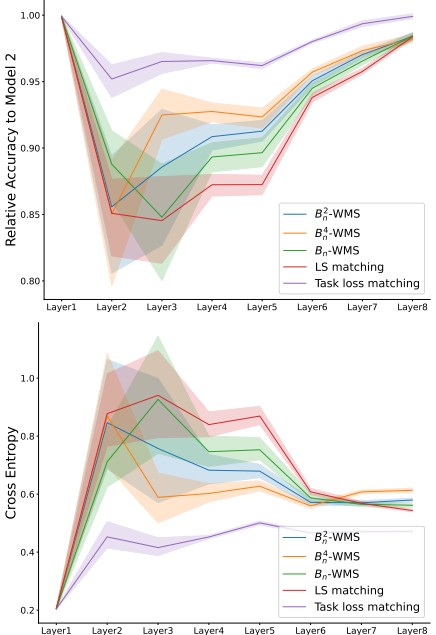

(a) Activation weighted mean squared direct matching (6), activations are raised on different exponents after normalization.

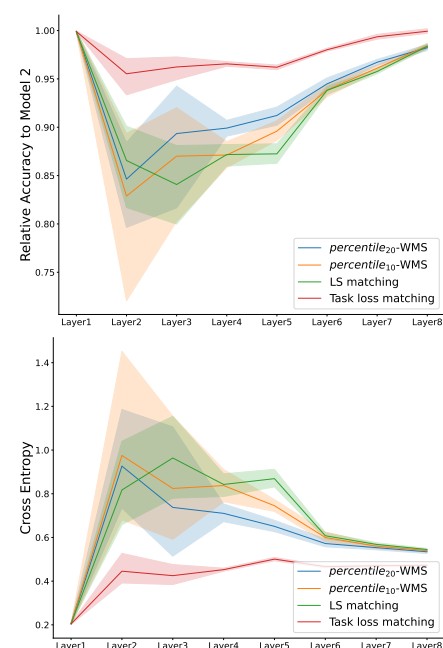

(b) 0-1 weighted mean squared direct matching (7), with the threshold $T_i$ set according to a percentile of the activations $b_{ij}$, and activations below the threshold are not matched.

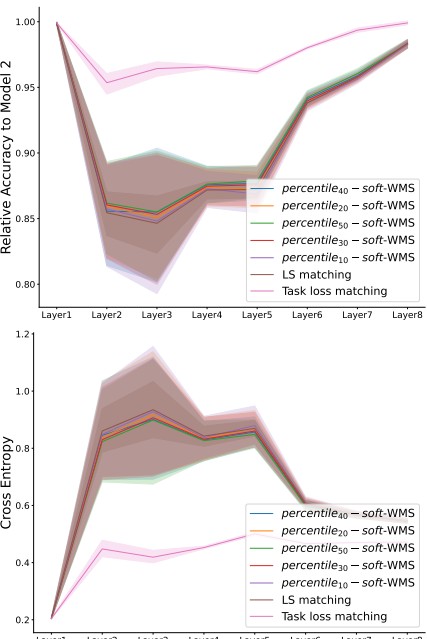

(c) Activation weighted mean squared direct matching (8), with the threshold $T_i$ set according to a percentile of the activations $b_{ij}$, and activations below threshold matched to stay below threshold.

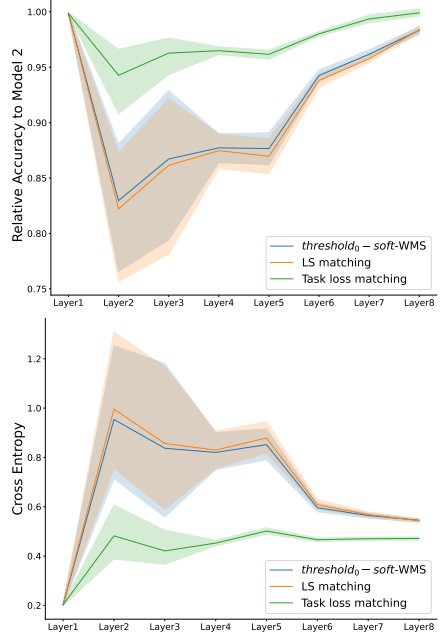

(d) 0-1 weighted mean squared direct matching (8), with the threshold $T$ set to 0, and activations below threshold matched to stay below threshold.

Figure 15: Relative accuracy to Model 2 and cross-entropy of Tiny10's stitched layers with different weighted mean squared direct matching methods.

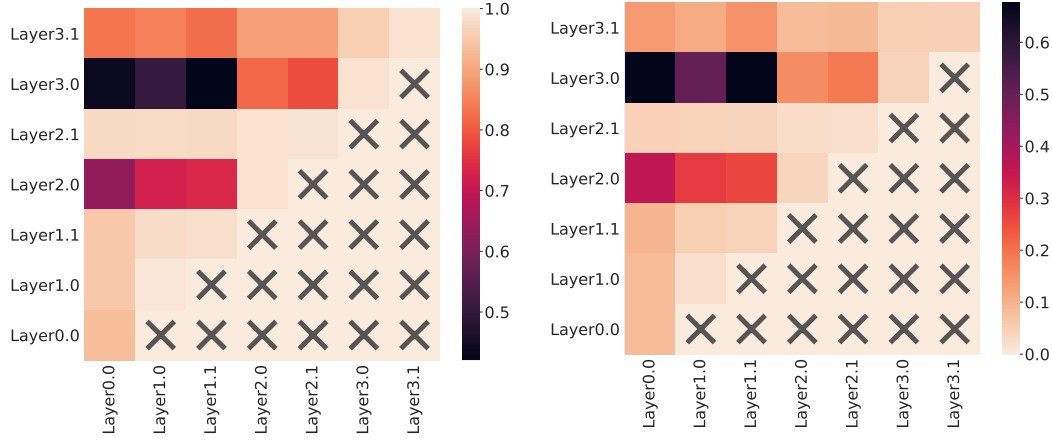

(a) Heatmap of relative accuracies to Model2.

(b) Heatmap of cross entropies between the stitched model's and Model2's outputs.

Figure 16: Stitching between two ResNet-20 networks, trained on the same data but with different sample order. The stitching layer was trained from random initialization. $i$th row and $j$th column refers to the two networks' $i$th and $j$th layers.

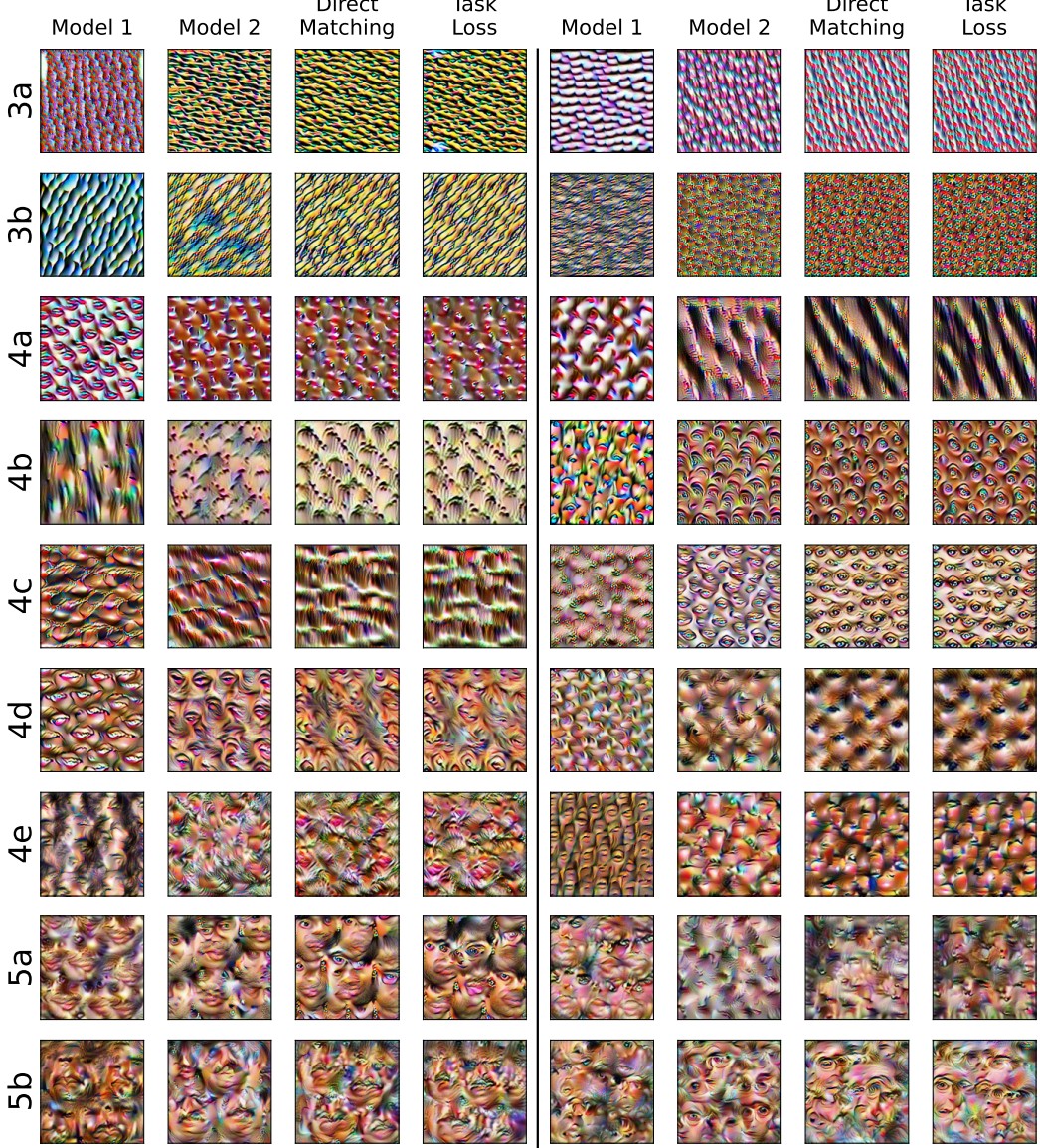

Figure 17: Stitching between two different CelebA models. All images are channel visualizations by Lucent. Rows correspond to layers where stitching and visualization happens. Leftmost four columns show first channel, rightmost four columns show second channel of the given layer. In each 4-column block, first column presents Model 1 channel, for comparison. Second column presents Model 2 channel, which in a sense is the target for the last two columns. Third column presents the channel linearly combined from Model 1 channels with the least squares error to Model 2 channel. (Direct matching.) Fourth column presents the substitute of the Model 2 channel created by stitching. Each row present the first two neuron of the corresponding layer.

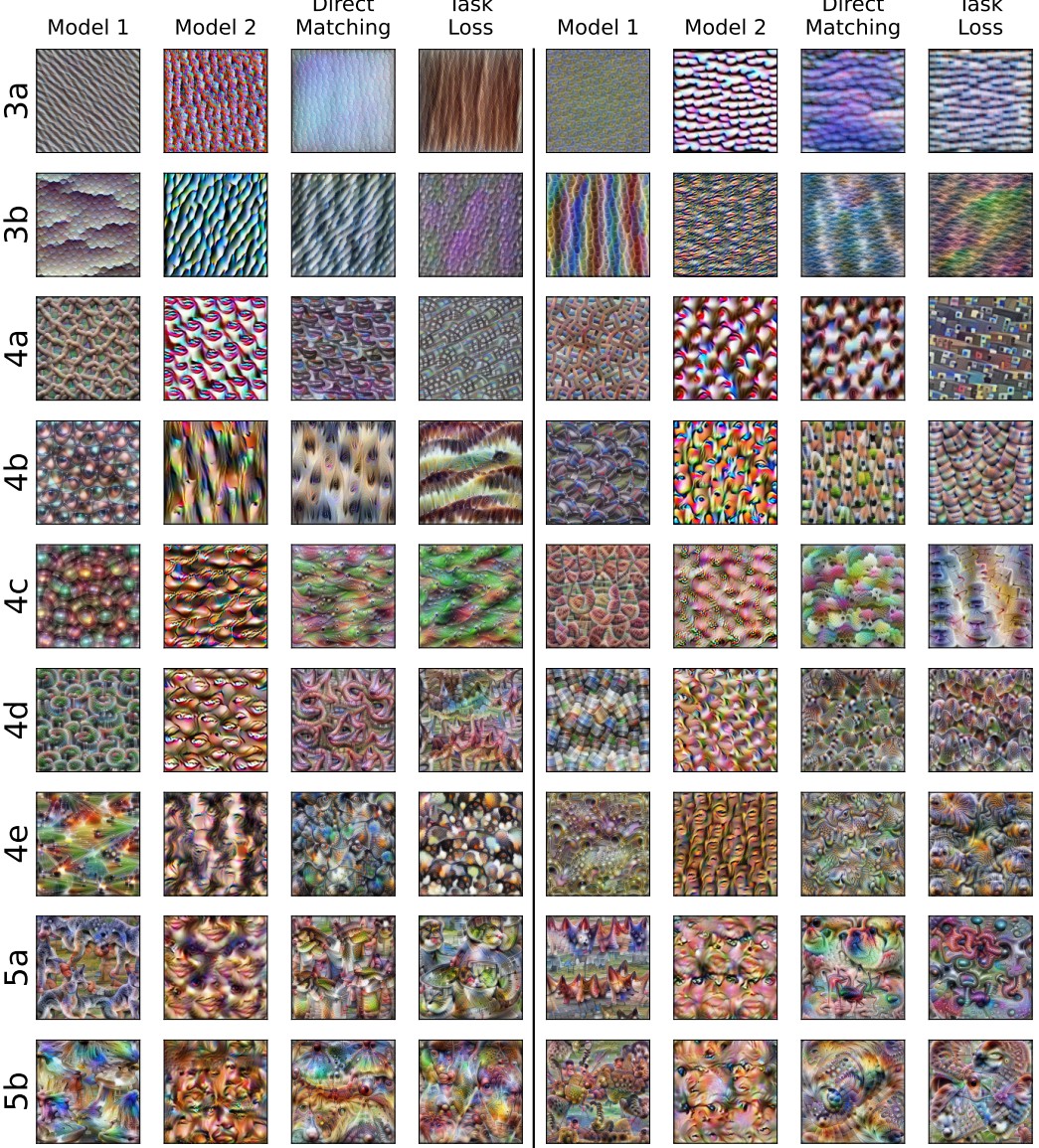

Figure 18: Stitching with Model 1 = ImageNet, Model 2 = CelebA. All images are channel visualizations by Lucent. Rows correspond to layers where stitching and visualization happens. Leftmost fours columns show first channel, rightmost four columns show second channel of the given layer. In each 4-column block, first column presents Model 1 channel, for comparison. Second column presents Model 2 channel, which in a sense is the target for the last two columns. Third column presents the channel linearly combined from Model 1 channels with the least squares error to Model 2 channel. (Direct matching.) Fourth column presents the substitute of the Model 2 channel created by stitching. Each row present the first two neuron of the corresponding layer.