# OpenReview forum: "Similarity and Matching of Neural Network Representations"
_NeurIPS.cc/2021/Conference — NeurIPS 2021 Poster_

### Official Review · Reviewer_yiYb · 2021-07-16

**Rating:** 7
**Confidence:** 3

**Summary:**

The paper proposes a new tool using a stitching layer to analyze the representational similarity between two deep neural networks (DNN). The idea behind the stitching layer is that it connects one frozen network to another via a parametrized layer that is optimized using the soft labels obtained from one of the frozen networks. The intuition is that if representations of two different networks are highly similar the final performance obtained using the stitching layer should be close enough to the original model's performance.

The key experiments demonstrate the following:
1. Using the stitching layer DNNs with the same architecture but different initialization can be matched almost perfectly
2. Standard similarity measures in the literature e.g. Centered Kernel Alignment(CKA) are not indicative of final task performance using stitching layer. Even with a low value of CKA, high performance can be obtained on the target task using stitching layers.


**Limitations And Societal Impact:**

Yes

**Main Review:**

**UPDATE: After reading the author's response and planned updates, I believe the paper will be valuable to researchers interested in applications of neural network similarity in functional usecases. Therefore I have updated my score to 7**

**Originality:** The paper introduces the idea of stitching layers to analyze similarities between neural network representations. The idea is derivative of [9] where they used a stitching layer to inspect equivalence between DNNs trained on different tasks. Here, the authors show results on comparing DNNs trained using different initializations.

The improvement on the original idea from [9] by using the optimal least-squares solution as initialization of the stitching layer is new. As shown in Figure 6, it improves the matching and therefore makes it possible to match even the deeper layers.

The comparison with CKA showing that the similarity measure is not indicative of task performance shares new insights into the limitations of these measures.

**Quality:**
The submission seems technically sound with well-designed experiments to support the key claims. The experiments showing the importance of OLS initialization (Experiment 5) and comparison with CKA (Experiment 3 and 4) are well designed. Further, the experiments are performed on three different architectures and two tasks showing the generalizability of the claims.

However, one important piece of information that is missing is which images were used to calculate CKA. If the number of images used to calculate CKA is less the results can be noisier, so it is crucial to provide these details.

Another weakness of the paper is that it shows the cases where the stitching layers work successfully. It might also be interesting to find out where they fail, for example, if two DNNs with the same architecture are trained on different tasks how successful is transformation using the stitching layer with respect to the depth of the network. How successful will the stitching be from a randomly initialized network?


**Clarity:**
The paper is well-written with the methods clearly explained through equations and Figures. The results section is also organized well.

**Significance:**
Identifying measures of similarity and their limitation is relevant to a wide set of applications such as transfer learning, multi-task learning, and interpreting what the network has learned. In this aspect, the paper introduces a new similarity measure based on task performance and shows the limitation of existing similarity measures.

However, the paper doesn't show where stitching layers fail to provide as good performance as the original models. Such analysis might have revealed new directions about the potential applications of the proposed measure.

**Time Spent Reviewing:**

6

---

> ### Author Response · Authors · 2021-08-10
> **response to reviewer yiYb**
>
> We would like to thank the reviewer for their valuable insights and comments on our manuscript.
>
> > However, one important piece of information that is missing is which images were used to calculate CKA. If the number of images used to calculate CKA is less the results can be noisier, so it is crucial to provide these details.
>
> CKA is calculated on the whole validation set, 10000 data points. We can fit this amount of data into memory without needing to resort to sampling methods like “minibatch CKA” [12]. We also note that the resulting CKA values are indistinguishable from values obtained when working with any number of data points between 2500 and 10000.
>
> > Another weakness of the paper is that it shows the cases where the stitching layers work successfully. It might also be interesting to find out where they fail, for example, if two DNNs with the same architecture are trained on different tasks how successful is transformation using the stitching layer with respect to the depth of the network. How successful will the stitching be from a randomly initialized network?
>
> Stitching between networks trained on different tasks is intended as follow-up work. We have already started experimenting with this setup. For example, we can report that a low-to-medium (below mixed4c) layers of an ImageNet inceptionV1 network can be stitched with the corresponding part of an inception network transferred from ImageNet to CelebA, without major loss in performance (92%->90%). Channel visualizations with the Lucid feature visualization framework demonstrate how the ImageNet channels organically adapt to their new CelebA task. In contrast, such stitching leads to major performance loss when done at the highest layers, an example of a case where stitching fails.
>
> > However, the paper doesn't show where stitching layers fail to provide as good performance as the original models. Such analysis might have revealed new directions about the potential applications of the proposed measure.
>
> We agree with the reviewer that presenting more settings where Frankenstein indicates dissimilarity would have been valuable. Indeed, our only such experiments involved additional constraints on the stitching layer (sparsity and rank constraints). This is mainly due to space limits. As noted in our response to Reviewer AJnQ, stitching between distinct layers of the same architecture, and stitching between layers of networks trained on different tasks gives many examples of dissimilarity.

---

> > ### Comment · Reviewer_yiYb · 2021-08-22
> > **Response to Author's response**
> >
> > I would like to thank the authors for their comprehensive response to my concerns as well as the concerns raised by other reviewers. My concerns have been adequately addressed.
> >
> > >For example, we can report that a low-to-medium (below mixed4c) layers of an ImageNet inceptionV1 network can be stitched with the corresponding part of an inception network transferred from ImageNet to CelebA, without major loss in performance (92%->90%). Channel visualizations with the Lucid feature visualization framework demonstrate how the ImageNet channels organically adapt to their new CelebA task. In contrast, such stitching leads to major performance loss when done at the highest layers, an example of a case where stitching fails.
> >
> > I encourage the authors to add the above results either in the paper or if the space does not allow it briefly report the above experiment in the paper and add the details in the supplementary. I strongly believe that reporting the above results will give readers more insights into the applicability of the stitching layers in future works.
> >
> > As mentioned by the other reviewers, some parts need more clarity in framing. If the authors include the above result and include clarifications as requested by other reviewers I recommend this paper for acceptance

---

### Official Review · Reviewer_PmP3 · 2021-07-17

**Rating:** 5
**Confidence:** 3

**Summary:**

The paper presents a framework to evaluate the similarity of intermediate representations from two neural networks. Given two neural networks with the same architecture but trained with different weight initializations, the framework uses a stitching layer to search for a transformation between the intermediate representations of the two networks, such that the model performance on a given task is preserved. The paper further proposed two approaches to obtain such transformations. Under this framework, the author empirically observed that representations from a given layer of two convolutional networks can be matched.

**Main Review:**

1. A core claim made by the paper is the empirical observation that under the proposed framework, representations from a given layer of two convolutional networks with the same architecture, but different weights can be matched. This claim is supported by the results in section 5 on three convolutional architectures (Tiny-10, ResNet-20, Inception-V1) on two tasks (CIFAR-10 and CelebA). However, it is not obvious whether the matching method is generic as claimed on line 100, even to other convolutional architectures. It is also unclear how architecture, tasks, training procedure can affect the observation. More explanation and descriptions should be made to this part.

2. I found the paper a bit hard to follow, especially the experiment section. As the proposed toolset is expected to provide an evaluation framework on representation matching, I would expect to see how the similarity metrics, namely the loss or accuracy when training with the stitching layer, is correlated with 1) matchability of the representations and 2) with the “ground truth” representation similarity. Yet the experiment section 5 is extensively about comparing different methods to obtain the stitching layer, while section 6 is only comparing with existing metrics (similarity indices).

**Time Spent Reviewing:**

3 hours

---

> ### Author Response · Authors · 2021-08-10
> **response to reviewer PmP3**
>
> We thank the reviewer for their valuable comments on our manuscript.
>
> > However, it is not obvious whether the matching method is generic as claimed on line 100, even to other convolutional architectures. It is also unclear how architecture, tasks, training procedure can affect the observation. More explanation and descriptions should be made to this part.
>
> We conducted experiments on many architectures (Tiny-10, Resnet with various widths, InceptionV1) and tasks (CIFAR-10, CelebA). We experimented with networks with batch normalization and without batch normalization, and also with networks which were trained with different optimizers (SGD, Adam). Still we did not encounter any cases where there were unmatchable instances.
> While this method is not universal on arbitrary architectures, this does not modify the observation that there exist stitching transformations in the wide diversity of the above cases. We found that the most sensitive part of the training process was initialization of the stitching layer, but we reviewed its effect on the training process on Figure 6.
>
> > As the proposed toolset is expected to provide an evaluation framework on representation matching, I would expect to see how the similarity metrics, namely the loss or accuracy when training with the stitching layer, is correlated with 1) matchability of the representations and 2) with the “ground truth” representation similarity.
>
> The correlation of matchability and “ground truth” representation similarity is exactly the content of our Experiment 3 (Figure 4).
>
> We will try to clarify the terminology that we used. We used matchability as an abstract notion, which we measured via the cross-entropy and relative accuracy of the stitched network. The abstract notion of representation similarity was measured via similarity indices, such as CKA and RLR.
>
> > Yet the experiment section 5 is extensively about comparing different methods to obtain the stitching layer, while section 6 is only comparing with existing metrics (similarity indices).
>
> The reason for examining these questions in Sections 5 and 6 is related to the first question of the reviewer, namely the dependence of the observations on the training procedure. Indeed, we found it important to deal with the stitching layer and the structure of its weights.
>
> We would like to emphasize that we did not intend to benchmark or compete with CKA or any other similarity measure. Instead, our goal was to provide a toolset with which we can observe, analyse and make statements about the relation of representational similarity and functional similarity. A relation that is highly relevant yet underexplored in the literature.

---

> > ### Comment · Reviewer_PmP3 · 2021-09-01
> > **Concerns can be addressed**
> >
> > Most of previous concerns can be addressed by the responses from authors. Therefore, I would like to increase the score from 5 to 6. In the future version, it will be great to add more explanations on the terminology used and also the relations to CKA methods.

---

### Official Review · Reviewer_AJnQ · 2021-07-18

**Rating:** 6
**Confidence:** 4

**Summary:**

**EDIT: I have increased my score from 4 to 6 based on the authors' responses and their proposed edits**

The submission considers the representational similarity between two neural networks by fitting linear transforms between corresponding layers. The authors investigate "direct mapping", where the linear transformation is learned simply based on the layer activations, and "task loss matching", where the linear transformation is fit to optimize task performance of the "Frankenstein" network (layers 1–n from model 1, layers n–end from Model 2), when holding all other network parameters fixed. They observe that a single linear 'stitching layer' can successfully map between corresponding layers of two networks of identical architecture and training, differing only in their random initialization. The performance of the Frankenstein network depends on the architecture, the fitting method (direct fit vs task loss matching), and the layer at which the stitching is performed. When tested on Inception, there is no difference between layers or fitting method. The dependence on fitting method and layer are most pronounced for ResNet-20. They compare their results to Centered Kernel Alignment to highlight that low CKA similarity can be associated with high classification accuracy. Additional experiments attempt to characterize the stitching layers.

**Limitations And Societal Impact:**

The authors briefly discuss limitations but do not mention any potential negative societal impacts. No potential negative societal impacts readily come to mind.

**Main Review:**

Some related work is missing. For example, the "Frankenstein networks" described here are quite similar to the "selfer" networks in Jason Yosinki's NeurIPS2014 paper: "How transferable are features in deep neural networks?". In that work, they similarly characterize the representations in neural networks by how interchangeable the heads/tails of the networks are, although without the addition of a stitching layer. Other work has also used layer-wise linear transforms to characterize representations, for example (Alain, G., & Bengio, Y. (2016). Understanding intermediate layers using linear classifier probes. ArXiv, 1610.01644v3). Romero 2014 used a least squares loss between the activations of two networks in work on distillation (Romero, A., Ballas, N., Kahou, S. E., Chassang, A., Gatta, C., and Bengio, Y. Fitnets: Hints for thin deep nets. arXiv preprint arXiv:1412.6550, 2014.) The submission does reference the Kornblith 2019 paper, but doesn't make explicit how the Dr Frankenstein framework presented here differs from the Linear Regression condition tested in that work. The authors could also consider pointing to work in neuroscience which compares layers of neural networks to regions along a sensory hierarchy in the brain via linear matching (e.g. Yamins, D. L. K., & DiCarlo, J. J. (2016). Using goal-driven deep learning models to understand sensory cortex. Nature Neuroscience, 19(3), 356–365. https://doi.org/10.1038/nn.4244).

I am unaware of another paper documenting the fact that one can learn a linear transformation between corresponding layers of two networks of identical architecture that preserves their performance. However, I disagree with the authors that this result is surprising or remarkable. They may want to elaborate on why this result is remarkable.

The submission focuses on the strengths of their work without carefully considering its weaknesses. The comparison to CKA feels somewhat incomplete. If the proposed framework is really intended to measure representational similarity, then it should be evaluated according to the same benchmark as CKA. Does Dr. Frankenstein find corresponding layers in two networks of identical architecture and training, differing only in their random initialization, to be most similar (relative to all other pair-wise comparisons)? Given its dependence on a linear transformation between layers, I suspect it will perform similarly to linear regression, which performed significantly worse than CKA on the layer matching task in Kornblith 2019. An honest discussion of the strengths and weaknesses would discuss when one might care more about task performance and when one might care more about the similarity of representational geometry.  Since CKA is only invariant to orthogonal transformations, not any invertible linear transformation, it is not surprising that the high performance matchings found here sometimes correspond to lower CKA scores. In fact, that is by design. Representational similarity is decidedly not just about task performance. One may obtain identical task performance with completely different features. One wants a similarity metric that is sensitive to those differences. However, I permit that there are certainly cases where task performance is what matters or where some insight about the network can be gleaned from analyses similar to the one presented here. But the submission does not present this as a tradeoff; they describe this characteristic of CKA purely as a weakness.

The submission is fairly well-written. However, I find that it somewhat over-complicates a relatively simple set of ideas.

The introduction could be improved by providing more motivation for why the authors chose to focus on matchability.

The most important contribution of this paper is the documentation of under what conditions one can arrive at a performance-preserving linear stitching layer between corresponding layers of two identical models (for the three models tested).  It is much less clear from the submission how the Dr. Frankenstein framework is useful or insightful as a method for measuring representational similarity.


**Time Spent Reviewing:**

3.25

---

> ### Author Response · Authors · 2021-08-10
> **Response to Reviewer AJnQ**
>
> Let us thank the reviewer for their time and their valuable comments, which we will implement in our revision.
>
>
> > **“Some related work is missing."**
>
> We are grateful to the reviewer for these additional references, and we will add these references to the related work.
>
>
> > **“The submission does reference the Kornblith 2019 paper, but doesn't make explicit how the Dr Frankenstein framework presented here differs from the Linear Regression condition tested in that work.”**
>
> There are several differences, let us mention a few:
>
> - In Kornblith et al. the optimal least squares matrix does not manifest as a transformation matrix (they compute the regression coefficient using a QR decomposition), whereas in our setting we explicitly determine these matrices, inspect and analyse them, and use them as the stitching layer in order to evaluate the second network.
>
> - We not just evaluate the performance of a stitched network using this matrix (direct matching) but also present and analyse a completely different method to train the stitched network from this initialization (task loss matching).
>
> - We work with 1x1 convolutions, thus our linear mappings are restricted to preserve the spatial structure of the activation map.
> Overall, our work introduces a new viewpoint of functional similarity to the world of similarity indices.
>
>
> > **“I am unaware of another paper documenting the fact that one can learn a linear transformation between corresponding layers of two networks of identical architecture that preserves their performance. However, I disagree with the authors that this result is surprising or remarkable. They may want to elaborate on why this result is remarkable.”**
>
> In our opinion, observing the sole fact in itself, that we are dealing with a deep, non-linear system and it is still possible to match representations with a single affine transformation highlights an important aspect regarding the inner workings of representation learning with neural networks. We are not aware of any theoretical work that would predict this phenomenon.
>
> Also, finding a good matching is not trivial: let us quote [22], which in our terminology is an instance of direct matching. The authors observe that their method fails to find good linear mappings for later convolutional layers:
>
> "The mapping layers for higher layers (conv3-conv5) showed poor performance even without regularization, for reasons we do not yet fully understand, so further results on those layers are not included here. Future investigation with different hyperparameters or different architectures (e.g. multiple hidden layers) could train more powerful predictive models for these layers."
>
> In contrast, in our work we achieve such mapping layers with least squares initialization, see Figures 6 and 9.
>
>
>
> > **“The submission focuses on the strengths of their work without carefully considering its weaknesses. The comparison to CKA feels somewhat incomplete. If the proposed framework is really intended to measure representational similarity, then it should be evaluated according to the same benchmark as CKA. Does Dr. Frankenstein find corresponding layers in two networks of identical architecture and training, differing only in their random initialization, to be most similar (relative to all other pair-wise comparisons)?”**
>
> We would like to emphasize that our framework is not intended to compete with CKA or any other similarity measure. Rather to provide a toolset with which we can observe, analyse and provide insights about the relation of representational similarity and functional similarity (through the notion of matchability) --- a relation that is highly relevant yet underexplored in the literature.
>
>
>
> > **"Given its dependence on a linear transformation between layers, I suspect it will perform similarly to linear regression, which performed significantly worse than CKA on the layer matching task in Kornblith 2019."**
>
> As per request of the reviewer, we did this experiment and found that it successfully identifies the blocks of the ResNet network with a block-diagonal heatmap. The short period of the rebuttal time did not allow us to do this full comparison for Tiny-10, because our method involves dealing with the spatial dimensions of the feature maps, and this is more problematic with Tiny-10.
>
>
> > **"An honest discussion of the strengths and weaknesses would discuss when one might care more about task performance and when one might care more about the similarity of representational geometry."**
>
> This is a central question of our work, even though we did not state it in terms of strengths and weaknesses. Namely, representational geometry is a widely investigated notion, which by definition does not depend in any way on the rest of the network. The performance on the task is not incorporated into such notions, to the best of our knowledge. A good notion of similarity - in our viewpoint - should directly incorporate information about later layers, for instance in a classification problem should depend on the classification boundaries. This is a central question, which we do not claim to answer, but which we also ask.
>
>
> > **"Since CKA is only invariant to orthogonal transformations, not any invertible linear transformation, it is not surprising that the high performance matchings found here sometimes correspond to lower CKA scores. In fact, that is by design. Representational similarity is decidedly not just about task performance."**
>
> We wholeheartedly agree that representational similarity is not just about task performance - we also attempted to indicate another variant of this idea on L237-242. There, we illustrate why identical representational geometry can achieve very different task performance. But the quoted experiment illustrates that different representational geometry on the same network can achieve the same task performance as the original. This captures an aspect of the flexibility of the information geometry with respect to a given task, which we found to be another explicit instance of the discrepancy between these two notions.
>
>
>
> > **"One may obtain identical task performance with completely different features. One wants a similarity metric that is sensitive to those differences. However, I permit that there are certainly cases where task performance is what matters or where some insight about the network can be gleaned from analyses similar to the one presented here."**
>
> This is an important observation - different features may produce identical task performance - it would be interesting to see whether Frankenstein would find such networks to be similar.
>
> However, let us emphasize that in our work we do not suggest replacing classical similarity measures with a single number produced by the stitched network, but rather suggest the viewpoint that a good notion of a similarity should also incorporate information about the performance on the task. To quote Kornblith et al: “Measuring similarity between the representations learned by neural networks is an ill-defined problem, since it is not entirely clear what of the representation a similarity index should focus on.” Our paper is hopefully a step in this direction.
>
> > **"But the submission does not present this as a tradeoff; they describe this characteristic of CKA purely as a weakness."**
>
> Part of our contribution is to emphasize that these similarities are not directly comparable and measure different aspects of neural representations. We attempted not to formulate this as a weakness, but rather point out certain phenomena that end-users might encounter (cf. Mirzadeh et al [12]).
>
> > **"The submission is fairly well-written. However, I find that it somewhat over-complicates a relatively simple set of ideas. The introduction could be improved by providing more motivation for why the authors chose to focus on matchability."**
>
> Indeed, some further discussion on the notion of matchability in the introduction would be warranted - due to the page limit, we had to be extremely succinct in several parts of the text. With the additional page, we will add a discussion and elaborate on the importance of functional similarity.
>
> > **"The most important contribution of this paper is the documentation of under what conditions one can arrive at a performance-preserving linear stitching layer between corresponding layers of two identical models (for the three models tested). It is much less clear from the submission how the Dr. Frankenstein framework is useful or insightful as a method for measuring representational similarity."**
>
> As the reviewer correctly points out, our framework is not suitable for measuring representational similarity. Instead, our framework gives a tool to measure functional similarity, and its relationship to representational similarity indices. Both of these viewpoints are valid, and in our opinion they should be considered together, as they capture different aspects of similarity.

---

> > ### Comment · Reviewer_AJnQ · 2021-09-01
> > **Concerns can be addressed with rewriting**
> >
> > Thank you to the authors for their detailed response. I agree with the other reviewers that with considerable rewriting of the text and reframing of the work, that that this paper can be suitable for publication.
> >
> > By 'representational similarity' in my review, I refer to any statements about the similarity of representations. I take this work to be concerned with representational similarity and the general question 'what does it mean for two representations to be similar?'. We can differentiate between methods that focus on representational geometry (CKA) or methods like those described in the present work which focus on what the authors call functional similarity. But in my mind, this is all still 'representational similarity' in that both these approaches are making characterizations about the similarity of pairs of representations, layers or networks. As you are paving new ground here, I urge you to be careful about the terminology you employ. It may be worth while to explicitly define these terms in the paper so that it is clear. If you wish to reserve 'representational similarity' for only statements about representational geometry, you can define it as such in your intro. However, I would be in favor of 'representational similarity' as a general term. Kudos to the authors for navigating this. It's clear that it is not straightforward to communicate precisely at this cutting edge but I hope that this review process has helped to make the paper more clear.
> >
> > The authors have shown that these two types of similarity can be in conflict. This is an important demonstration. A productive discussion will describe when one might prefer similarity functions based on representational geometry, functional similarity, or some combination of the two. I think for many applications that currently use representational geometry, incorporation of functional similarity will not be necessary or helpful. As such, I would hope for a more specific description of the cases where functional similarity will be relevant to researchers or practitioners rather than blanket statements. If the authors wish to argue that all similarity analyses should incorporate functional measures, that would be a much more difficult claim to support.
> >
> > With the edits proposed by the reviewers and accepted by the authors, I will increase my score to a 6 and recommend acceptance.

---

### Official Review · Reviewer_j6kV · 2021-07-25

**Rating:** 7
**Confidence:** 3

**Summary:**

The paper develops and applies a set of methods to determine the functional equivalence of architecturally-identical neural networks trained with different initializations. These methods involve taking the activations from a given layer l in one network, inputting them into a "stitching layer" (either a linear projection or a 1x1 convolutional layer), then taking the activations from the stitching layer and inputting them into layer l+1 in a second network. The stitching layer is optimized either to minimize the models' loss on a task, or to minimize the distance between its own outputs and the activations of the second network. The authors find that stitching layers can typically be learned that have minimal impact on model accuracy and loss. They also demonstrate some results that they claim undermine the utility of CKA, a popular representational similarity metric.

**Limitations And Societal Impact:**

The authors only provide three sentences on the limitations of their work. I understand space is limited, but I think they could acknowledge the limited size of the datasets they use, and some of the instances in which their results were inconsistent and/or difficult to interpret.

**Main Review:**

**UPDATE: I have increased my rating from a 3 to 7 in response to the authors' responses**

I think the approach presented in this paper is clever and relevant, and could yield results that are worthy of publication. But I have a hard time feeling confident about the work in its current form. It’s difficult to assess whether the results are meaningful, both because of a lack of clear baselines and controls, and a lack of interpretation and explanation by the authors. The results are also limited to very small models and datasets. However, I am optimistic that my concerns can all be addressed through additional analyses and writing.

I applaud the authors’ use of replicates and measures of variability in plots and experiments. This is necessary but disappointingly uncommon in empirical machine learning research.

The authors open with the question “When are two representations the same?”. This statement could be refined. “The same” is a binary assessment, but the paper seems interested in measures of similarity, and similarity is a continuous quantity. I think the present work attempts to dissociate functional similarity (e.g. changes in performance measures) and representational similarity, and the authors should focus on this distinction. I don’t mean to be pedantic—I think the work is well-motivated and the problem is relevant, but the paper opens with a framing of the problem that undersells it.

One general issue I have with this paper (which recurs in some of the comments below) is a lack of clear baselines for establishing whether some change in a quantity is meaningful. For example, what is the cross-entropy between model outputs at init? Or when the stitching method is applied to two networks at init? Or when the task loss/stitching layer method is applied, but the stitching layer is trained to minimize the *change* in the model’s outputs before vs. after stitching? All of these would help contextualize the current results.

The authors only compare their results to CKA, choosing not to use CCA-based representational similarity measures because those “indices do not succeed in identifying similar layers of identical networks trained from different initializations” [L67-68]. Just because CCA-based methods are more sensitive to differences in initialization than CCA doesn’t mean they’re irrelevant to the present work. The present work develops methods to test whether differences in initialization are meaningful, so it seems sensible to me to include other representational similarity methods as a control.

One way the authors demonstrate the potential unreliability of CKA is by plotting raw CKA scores vs. cross-entropy (Figures 4 and 5). This is difficult to interpret without some baseline or context for what magnitude change in these quantities is meaningful. It also needs to be quantified. At the very least, compute a non-parametric correlation between these quantities. This extends to other representational similarity metrics, which I would encourage using (as per the previous comment).

The authors also demonstrate the unreliability of CKA by “stitching in” a linear layer initialized with an identity mapping, then training the layer using the task cross-entropy and a term to penalize the CKA value at the stitch. They find that they are able to maintain high accuracy while causing the CKA value to decrease. I am skeptical of this result for multiple reasons. Related to my previous comment, there is no context to determine whether this change in CKA value is meaningful (it even looks like accuracy decreases a small amount!), and the result is not quantified. Furthermore, CKA was specifically designed not to be invariant to all invertible linear transformations. Given the small changes in accuracy and cross-entropy, I suspect that the stitched-in layer simply learned a transformation to which the network is *less* sensitive than CKA.

It’s not clear to me what the authors mean when they state that they initialize from the optimal least squares matching (e.g. Line 180). Do they mean that the activations from Network 1 are first transformed using the linear projection obtained by solving a least squares mapping between Networks 1 and 2, and then the linearly-projected activations are fed into the convolutional layer?

I think it might be helpful to attempt to dissociate the effect of the type of projection used in the stitching layer from the effect of the objective. Is it possible to solve the linear projection(s) using the task loss as the objective?

Is it possible to use the objectives of representational similarity measures (e.g. CKA, CCA) for direct matching? If so, it could demonstrate the Frankenstein framework’s utility for benchmarking representational similarity measures.

It’s not clear to me how to interpret the sparsity results. The section is titled “​​direct vs. task loss matching”, but these conditions are not directly compared in the text, and it’s not immediately apparent to me that this comparison is meaningful. Is there a meaningful baseline that could be used here? Perhaps inserting a linear layer at the same depth as the stitching layer, then training the layer to maximize both sparsity and accuracy, and examining how accuracy changes as a function of sparsity?

Some of the results are very interesting, but nearly all of them are presented without much interpretation. For example, what is the implication that Procrustes matching yields similar results as linear least squares matching for later layers but not for early layers [L175-177]? What is the implication of the lack of mode connectivity for stitching layer solutions in early layers?

What is Layer 0 in the Resnet-20 (Figure 2) and why is performance so low when using it to stitch/match representations?

What models are used in Figures 4-8?

In Figure 8b, the x-axis is labeled “sparsity”. How is this computed?


**Time Spent Reviewing:**

4 hours

---

> ### Author Response · Authors · 2021-08-10
> **Response to Reviewer j6kV**
>
> We thank the reviewer for their thorough review and their constructive input to our manuscript. First, let us comment on two of the main concerns of the reviewer.
>
> The first issue concerned the relationship of this work to CKA. This work does not intend to question the utility of CKA - this has been clearly demonstrated in several papers. Our goal was rather to provide an alternative viewpoint on similarity: as the reviewer aptly put it, functional similarity. This is a relatively under-examined notion in the literature, even though the field is largely concerned with performance of neural models on different tasks, which emphasizes the importance of the functional perspective. In order to explore the relationship between these two viewpoints, we chose the most advanced incarnation of representational similarity indices, CKA. Our toolbox introduces the possibility to compare similarity indices and their relation to functional similarity.
>
> The second main issue concerned missing baselines. We thank the reviewer for drawing our attention to these. Below we address this concern by providing such baselines for each of our main experiments, and give interpretations further below. In many cases, these baselines are implicitly present, e.g., relative accuracy in Figure 2. We will adjust the camera-ready version accordingly. Here is a brief summary of the baselines for the quantities:
>
> - Relative accuracy (Experiments 1, 2, 4, 5 - Figures 2, 3, 5, 6): Relative accuracy is already computed with respect to a baseline, namely the performance of Model 2 on the given task.
>
> - Cross-entropy (Experiments 1, 3, 4 - Figures 2, 4, 5): We agree with the reviewer that raw cross-entropy values without a baseline are hard to interpret. We have remedied this and provide the following baselines:
>    - Average cross-entropy between Model1 and Model2 outputs: Tiny-10: 0.061 (std: 0.002), Resnet: 0.047 (std: 0.001)
>    - Average cross-entropy between Model2 outputs and the ground truth labels: Tiny10: 0.460 (std: 0.011), Resnet: 0.387 (std: 0.006)
>    - Average cross-entropy between Model1 and Model2 when the transformation is still not trained: Tiny-10 (Layer 3): 0.8909 (std: 0.1765). (While this depends on the layer of the stitching, as an untrained stitching layer basically results in random outputs therefore this could serve as a reference value.)
>
> - CKA (Experiments 3, 4 - Figures 4, 5): To provide a CKA value which represents a good matching, we consider the following baseline: the CKA values between representations of the transformed activations of Model1 and Model2 averaged over 10 different network pairs and all layers. The stitching layer was trained with the task loss for these baselines.
>    - ResNet: 0.922 (+-0.098)
>    - Tiny-10: 0.883 (+-0.050)
> We also refer to further baseline CKA values to Kornblith et al. [8, Figure 2].
>
> - Relative accuracy w.r.t. rank (Experiment 6 - Figure 7): Here the baseline is the full rank case, the performance change of the low rank cases can be compared to this. This is contained in the plot.
>
> - Relative accuracy w.r.t. sparsity (Experiment 7 - Figure 8): The baseline is the 0 sparsity case, which is already on the plot.
>
> We thank the reviewer for drawing our attention to these and thereby contributing to an improved version of our paper.
>
> **We are pleased that in summary the reviewer found our approach to be valuable and we hope to clear up most of the remaining concerns below. We will now address these one by one.**
>
> > **“The authors open with the question “When are two representations the same?”.”**
>
> We agree with the reviewer that this rhetorical question introducing the topic of the paper is likely an ill-posed one, replacing “same” with “similar” will be a better suited word here.
>
>
> > **“One general issue I have with this paper (which recurs in some of the comments below) is a lack of clear baselines for establishing whether some change in a quantity is meaningful. For example, what is the cross-entropy between model outputs at init? Or when the stitching method is applied to two networks at init? Or when the task loss/stitching layer method is applied, but the stitching layer is trained to minimize the change in the model’s outputs before vs. after stitching? All of these would help contextualize the current results.”**
>
> We partially addressed this question of baselines above, let us provide some additional details.
>
> We thank the reviewer for the first two suggestions, which are included above, and we now consider these as baselines.
>
> Regarding the third suggestion, if we understand the reviewer correctly, that is exactly how we conducted our experiments: the loss term for the task loss was the cross-entropy between the output of the stitched network and the original outputs of Model2. We also experimented with utilizing the cross-entropy between the output of the stitched network and the ground truth labels. The results were basically the same, but we found the former choice to be philosophically more grounded. (We clarify this in the text in lines 117-123.)
>
>
> **Regarding CCA-based measures**
>
> In fact we did do experiments with CCA, namely Yanai’s GCD measure, however we found that in our main setting - comparing representations obtained from different random initializations - it produced low values on representational similarity. This is in line with the findings of Kornblith et al [8]. Since our experiments also involved evaluating similarity indices on networks trained from different initializations, we chose to simplify our presentation by concentrating our efforts on CKA (and also RLR). However, we agree with the reviewer that the reason for this omission should be mentioned in the paper.
>
>
> **On the CKA experiments**
>
> We addressed the question on the baselines above. We agree that some further interpretation would be beneficial here, so let us address these issues.
>
> When two Tiny-10 networks are trained on different initializations, CKA between layers N and N+3 is approximately 0.7, see Kornblith et. al [8, Figure 2].
>
> We also agree with the reviewers interpretation: the found stitching layer must be a linear transformation to which accuracy is less sensitive than CKA. Let us remark that it is our toolset that makes such a statement on the sensitivity particularly easy to state and test, which supports the versatility of Frankenstein.
>
> CKA vs accuracy: We do not intend to present this experiment as an unreliability of CKA, but rather as another instance where there is a discrepancy between functional and representational similarity. For instance, this discrepancy was noticed and briefly commented in [12, Mirzadeh et al. page 2], and we hope that our viewpoint clarifies the reasons.
>
> Decrease in accuracy: Since the stitching network in this experiment is trained on exactly the same network, the initial value of relative accuracy is 1.0, from which there can only be a decrease.
>
> Regarding Figure 4: In this experiment, we found that throughout the training process, as we start from the least squares initialization, the cross-entropy decreases and relative accuracy increases, while --- in a sense contrary --- the CKA values are increasing, thus the improvement in functional similarity happens against the representational similarity.
>
>
>
> > **“It’s not clear to me what the authors mean when they state that they initialize from the optimal least squares matching (e.g. Line 180). Do they mean that the activations from Network 1 are first transformed using the linear projection obtained by solving a least squares mapping between Networks 1 and 2, and then the linearly-projected activations are fed into the convolutional layer?”**
>
> Yes, this is exactly the case. We have made clarifications in the text to better reflect this.
>
>
> > **“I think it might be helpful to attempt to dissociate the effect of the type of projection used in the stitching layer from the effect of the objective. Is it possible to solve the linear projection(s) using the task loss as the objective?”**
>
> We are not sure that we understand this question correctly. In our interpretation, task loss matching is doing exactly what the reviewer asks for here. The task loss is not a linear function of the stitching weights, hence we optimize the stitching weights using gradient descent, starting from the aforementioned least squares initialization.
>
>
> **CKA or CCA as a direct matching method**
>
> CKA is not directly suitable as an objective for direct matching, since it is invariant to orthogonal transformations. A stitching with an optimal CKA value can be arbitrarily transformed via orthogonal transformations without affecting CKA, but most of these equivalent solutions are unsuitable for the task.
>
> Notions for representational similarity and functional similarity have different motivations, and thus, aim to capture different invariances. These invariances could be incompatible as in the above case of orthogonal invariance.

---

> > ### Comment · Reviewer_j6kV · 2021-08-20
> > **Response to authors' response**
> >
> > I really appreciate the thorough, detailed response. A number of my concerns were very clearly the result of misunderstanding or misinterpretation, and I thank the authors for their patient and meticulous explanation. The need for this explanation could arguably be a critique on the clarity of the work, but it is my hope that the extra page provided for revised manuscripts would afford this clarity. I also hope the extra page accommodates the interpretation and discussion of the findings and their implications. The authors provided some of this in their response and it was helpful and interesting. The baselines—both those that are new and those that I missed—make it much easier to interpret results.
> >
> > In reading the other reviewers' reviews and rereading my own, it's clear this work has an issue with framing (or at least perceived framing): three out of four of the authors' responses included a statement clarifying that they were not intending to benchmark/compete with/quantify the efficacy of CKA, and/or that Dr. Frankenstein is for analyzing functional similarity, not representational similarity. An unacceptable solution would be to simply state these facts in the manuscript. A comprehensive solution would directly and thoroughly explicate the conceptual framework underlying this work. The authors should lay out the distinction between functional and representational similarity in the introduction. What do functional similarity measures capture that representational similarity measures don't? Why should we care about functional similarity? Are there existing methods for quantifying functional similarity? This last question is important, and potentially demands the inclusion of suitably similar methods as a baseline/control, or a good justification for not including them should they exist.
> >
> > So long as the authors update their manuscript to include the clarifications and amendments in their responses here, and they add a clear articulation of functional vs. representational similarity, I would be happily endorse this paper for acceptance.

---

> ### Author Response · Authors · 2021-08-10
> **Continuing the response for Reviewer j6kV**
>
> [Continuing our response for Reviewer j6kV]
>
> > **"It’s not clear to me how to interpret the sparsity results. The section is titled "direct vs. task loss matching", but these conditions are not directly compared in the text, and it’s not immediately apparent to me that this comparison is meaningful. Is there a meaningful baseline that could be used here? Perhaps inserting a linear layer at the same depth as the stitching layer, then training the layer to maximize both sparsity and accuracy, and examining how accuracy changes as a function of sparsity?"**
>
> Indeed, in our paper, we do exactly as the reviewer suggests. In these experiments, both task loss matching and least squares direct matching are regularized the same way to achieve sparsity: utilizing an L1-regularization loss term. Accordingly, here the least squares direct matching is also solved with gradient descent (with MSE on the activations as the main loss term) instead of an analytical solution.
>
> Sparsity baseline: the dots with sparsity around zero in the scatter plot of Figure 8 correspond to the unregularised baseline (here the L1-term equals zero). These sparsity numbers measure the sparsity of the stitching layer, hence here the 0 sparsity case should be considered as a baseline for investigating the dependence of accuracy on sparsity.
>
> Sparsity interpretation: Some brief comparisons of direct matching and task loss matching can be found in the caption of Figure 8 and lines 290-291. Additionally, we find that analysing sparsity in this context provides interesting insights, e.g., with regards to  [10, Li et. al.], see L53-58. Our experiment also demonstrates that even though high accuracy sparse matchings might not be produced by direct matching, it is possible that there still exists a meaningful alignment of the neurons using additional information about the network. Solutions with sparse transformation matrices correspond to matchings with more aligned neurons (e.g., in the extreme case of a permutation transformation matrix, there would be a one-by-one correspondence between the neurons of the networks).
>
> > **“Some of the results are very interesting, but nearly all of them are presented without much interpretation.”**
>
> Indeed, due to the page limit, we had to be extremely succinct in several parts of the text. With the additional page, we can be clearer and elaborate further on these points.
>
>
> **Regarding Procrustes**
>
> The reason for the described phenomenon is not entirely clear to us. When looking at the general linear stitching at some later layer, the singular values of the transformation matrix span several orders of magnitudes. Hence it is not an a priori property of such stitchings that they are close to being orthogonal.
>
>
> > **“What is the implication of the lack of mode connectivity for stitching layer solutions in early layers?”**
>
> One reason for this might be that the representations emerging in early layers could be more diverse across networks trained from different initializations. As a consequence, linear interpolation might fail to produce meaningful transformation matrices.
>
> > **“What is Layer 0 in the Resnet-20 (Figure 2) and why is performance so low when using it to stitch/match representations?”**
>
> This is detailed in the supplementary material, Appendix A.2 line 31-32. Layer 0 denotes Layer 0.0 (we will make a correction here in the text), which corresponds to the activation space after the first Conv-Batchnorm layer in the network preceding the residual blocks.
>
> > **“What models are used in Figures 4-8?”**
>
> These can be generally found either in the captions or the supplementary material, for which the corresponding part is referenced in the main text. For convenience, we collect this information here.
> Figure 4: Tiny-10, stitching is at Layer 3.
> Figure 5: Resnet with 3x width, stitching is at Layer 2.0. (This was indeed missing from the text, we added it.)
> Figure 6, 7, 8: Tiny-10, stitchings are on basically all layers (appendix contains more figures).
>
> > **“In Figure 8b, the x-axis is labeled “sparsity”. How is this computed?”**
>
> We set every entry of the stitching matrix S to zero when the absolute value of the entry is below the threshold 10^-4, and compute the ratio: sparsity = (nonzero elements in S) / (total number of elements in S).

---

### Author Response · Authors · 2021-08-24
**Summary of revisions in light of reviews**

We are delighted to hear from both Reviewer j6kV and Reviewer yiYb that their concerns have been adequately addressed. We agree that the framing of the Introduction could be improved upon and we welcome the Reviewers’ intention to endorse our paper provided we add the clarifications described in our responses. As no paper revisions may be submitted during the review process, below we list the changes that we have made or plan to make for the camera ready version based on the discussions and the requests of the Reviewers to date. We hope that these modifications also address the concerns of Reviewer PmP3 and AJnQ, but naturally we are open to any further discussion and to address any possible concerns. We hope that with these changes we meet the Reviewers’ expectations for the acceptance of the paper.


We plan to extend the introduction to outline a conceptual framework that unfolds the subjects of the paper more naturally. This would encompass:
- The discussion of representational and functional similarity.
- The different motivations of these two viewpoints on similarity, the compatibilities and possible inherent incompatibilities of these notions. The benefits of investigating these notions together.
- The significance of functional similarity of neural network components and its relation to other techniques, such as model distillation, teacher-student training schemes or transfer learning.
- The novelty of investigating representational similarity measures from this viewpoint and the lack of any similar methodology in the literature.
- When interpreting our results we will refer to this framing to enhance the coherence of the text.

We will implement the following amendments, each of which already have been addressed in our responses in detail:

- We detail the baselines for the cross-entropy values in the section for Experiment 1, and also put them on Figure 2 and Figure 4.
- We report and discuss the reference CKA values of the well performing stitchings and refer for further baseline CKA values to Kornblith et al. in the text of Experiment 3 and 4.
- We will introduce a paragraph in Section 6 that briefly discusses our results with CCA.
- We will add a paragraph to Section 6 that discusses the possibility (in fact, the inadequateness) of the direct usage of CKA as an optimization objective for finding matchings.
- We will add a small section in the main text that discusses our experiments regarding stitching between corresponding layers of networks trained on different data.
- We will include additional content in the supplementary material with Lucid feature visualizations depicting how the stitching reuses Imagenet features to construct CelebA features.
- We will further elaborate on the interpretation regarding the sparsity experiments.
- We will elaborate on the relationship of the Dr. Frankenstein framework to the Linear Regression condition tested in Kornblith et al., as recommended by Reviewer AJnQ.
- In the supplementary material we will include the layer-to-layer matching performance heatmap provided by Dr. Frankenstein as suggested by Reviewer AJnQ.
- We extend the paragraph in Experiment 1 detailing the wide diversity of neural network configurations where we --- nearly without exception --- successfully matched neural representations (in the light of the response of Reviewer PmP3). Here we will also highlight the rare cases where our methods failed to find a well-performing stitching. (As per request of Reviewer yiYb.)

Smaller corrections:

- We will add further details regarding the calculation of the CKA measure (as recommended by Reviewer yiYb).
- We will include additional references as suggested by Reviewer AJnQ.
- We changed the word “same” to the more adequate “similar” in the rhetorical question of line 17.
- We have made clarifications in the text regarding the optimal least squares initialization (after line 173 in current version).
- We have fixed the typo in Appendix A.2 line 31-32 regarding Layer 0.
- On Figure 5 we will add that the network was Resnet with 3x width and stitching was done at Layer 2.0.
- In Experiment 6 and 7 the adequate baselines are already there, but we will also reflect on this in the text.

---

> ### Comment · Reviewer_j6kV · 2021-08-24
> **Satisfactory revisions**
>
> I am quite satisfied with this summary of revisions. I believe a well-articulated framework examining functional vs. representational similarity will be valuable to the field in its own right. I look forward to reading the revised manuscript. I have updated my score from a 3 to a 7.

---

### Decision · Program_Chairs · 2021-09-27

**Decision:**

Accept (Poster)

**Comment:**

This paper examines performance when an affine layer is used to “stitch” together layers from two neural networks, so that the hidden representations of one network are used as inputs to an intermediate layer of the other. Reviewers agreed that study of performance achievable via matching with either least squares or task loss was an important contribution. However, several reviewers expressed concern regarding the framing of the work, and in particular, the lack of clarity regarding the relationship between representational similarity methods and the proposed framework. At the urging of Reviewer j6kV, the authors proposed to change the introduction to describe the relationship between representational and functional similarity and to motivate the work from this perspective. Reviewers also raised several more technical questions. The authors provided a summary of the revisions they plan to make, and after reviewing this summary, all reviewers have recommended acceptance. The AC agrees that the paper, with the proposed revisions, provides a valuable perspective on representational and functional properties of neural networks.